# Multiclass Boosting and the Cost of Weak Learning

**Nataly Brukhim**
Princeton University
Google AI Princeton
nbrukhim@princeton.edu

**Elad Hazan**
Princeton University
Google AI Princeton
ehazan@princeton.edu

**Shay Moran**
Technion
Google Research
smoran@technion.ac.il

**Indraneel Mukherjee**
indraneel.mukherjee@protonmail.com

**Robert E. Schapire**
Microsoft Research
schapire@microsoft.com

## Abstract

Boosting is an algorithmic approach which is based on the idea of combining weak and moderately inaccurate hypotheses to a strong and accurate one. In this work we study multiclass boosting with a possibly large number of classes or categories. Multiclass boosting can be formulated in various ways. Here, we focus on an especially natural formulation in which the weak hypotheses are assumed to belong to an "easy-to-learn" base class, and the weak learner is an agnostic PAC learner for that class with respect to the standard classification loss. This is in contrast with other, more complicated losses as have often been considered in the past. The goal of the overall boosting algorithm is then to learn a combination of weak hypotheses by repeatedly calling the weak learner.

We study the resources required for boosting, especially how they depend on the number of classes $k$, for both the booster and weak learner. We find that the boosting algorithm itself only requires $O(\log k)$ samples, as we show by analyzing a variant of AdaBoost for our setting. In stark contrast, assuming typical limits on the number of weak-learner calls, we prove that the number of samples required by a weak learner is at least polynomial in $k$, exponentially more than the number of samples needed by the booster. Alternatively, we prove that the weak learner's accuracy parameter must be smaller than an inverse polynomial in $k$, showing that the returned weak hypotheses must be nearly the best in their class when $k$ is large. We also prove a trade-off between number of oracle calls and the resources required of the weak learner, meaning that the fewer calls to the weak learner the more that is demanded on each call.

## 1 Introduction

Boosting [16] is a fundamental primitive in machine learning, most widely studied and applied in the context of binary supervised learning. Yet many important problems require classification into a great many target classes. For example, in image object recognition and building language models, the number of classes scales as the number of possible objects, or the dictionary size, respectively. It is thus of practical as well as theoretical interest to generalize boosting to the multiclass setting.

35th Conference on Neural Information Processing Systems (NeurIPS 2021).

In contrast with binary classification, there are several versions of multiclass boosting since there is more than one way to extend the weak learnability assumption. With only two labels, we simply require that the weak learner find a hypothesis with error slightly better than $1/2$, that is, a bit better than random guessing. So when the number of labels $k$ is more than 2, perhaps the most natural extension requires that the weak learner outputs hypotheses with classification error slightly better than a random guess among the $k$ labels, that is, with error a little better than $1 - 1/k$. However, this turns out to be too weak for boosting to be possible. Instead requiring classification error less than $1/2$ is sufficient for boosting, but far stronger than mere random guessing. Other approaches are based on reducing the multiclass problem to multiple binary problems, or constructing loss functions to be minimized by the weak learner that are significantly different than the simple classification error. See [13, 16] and references therein for a detailed discussion.

**A Weak Learning Assumption.** In this work we return to the classical formulation of boosting in which the weak learner aims to minimize ordinary classification loss, though in a way that sidesteps some of the hurdles just discussed. Specifically, we assume the weak learner is an agnostic PAC learner for some base class, meaning that it can find the approximately best hypothesis in that class. This requirement differs subtly, but importantly, from the kind we have been discussing in which, say, the weak hypotheses must have error not greater than some value; instead, the criterion is relative to this particular base class. Further, we take this class to consist of *simple* hypotheses, in line with the common supposition that weak hypotheses are rules of thumb from an "easy-to-learn" class [16, 18]. Thus, the weakness of the weak learner is manifested in the simplicity of the hypotheses in the base class, and the goal of the overall boosting algorithm is to learn target concepts *outside* the base class by aggregating the weak hypotheses provided by the weak learner. A similar setting was recently considered by [3] who studied binary-labeled classification.

Which target concepts can be approximated by boosting weak hypotheses from a given base-class? In this work we assume that the target concept can be represented by *weighted plurality votes* over the base class. To motivate this assumption, note that: (i) Most boosting algorithms (in both binary and multiclass settings) output a weighted vote of weak hypotheses [16, 10, 3, 8, 1]. Thus, if one focuses on such algorithms, it becomes a *necessary* condition that the target concept can be approximated arbitrarily well by such votes, and hence yields our assumption. (ii) In addition, the work by [16] conducted a thorough study of a variety of natural weak learning assumptions for multiclass boosting, and showed that each of those assumptions *implies* the plurality vote assumption. (iii) Finally, weighted votes are extremely expressive: for example, in the binary-labelled case, the work by [3] showed that under extremely mild assumptions on the base-class, one can approximate arbitrarily complex concepts by weighted votes. Relatedly, [19] proved that assuming a non-trivial base-class, one can obtain Bayes-consistent learning algorithms by boosting using weighted votes.

## 1.1 Contributions

We study the resources required for boosting, especially how they depend on $k$, the number of classes. For the booster, these include the number of examples needed for learning and the number of oracle calls to the weak learner. For the weak learner, we measure the required accuracy, that is, how close the returned hypothesis must be to the best in the base class, or alternatively, the number of unweighted samples the boosting algorithm feeds the weak learner in each round.

**A Multiclass Boosting Algorithm.** Our starting point (and first contribution) is a natural variant of AdaBoost (Algorithm 1), which demonstrates that it is possible to replace previously considered weak learners by one which aims to minimize the classification loss. Indeed, we prove that Algorithm 1 is able to learn any task which satisfies the general multiclass weak learning assumption of [13], which they proved to be necessary and sufficient (see Section 2).

However, a closer inspection of our algorithm reveals a surprising phenomenon: while the sample complexity of the booster exhibits a standard logarithmic dependence on $k$ (the number of labels), the sample-complexity of the weak learner depends quadratically on $k$, which is *exponentially* worse. In more detail, our algorithm requires that at each round $t$, the weak learner returns a hypothesis $h_t$ whose excess loss relative to the best hypothesis in the base class is less than $1/k$, which means, since $k$ is assumed to be large, that $h_t$ is nearly optimal. Stated differently, using standard statistical arguments, in each round $t$, our boosting algorithm provides the weak learner with $O(k^2)$ unweighted training examples (ignoring parameters other than $k$). This demonstrates a striking discrepancy

between the amount of information (in the form of labeled examples) the boosting algorithm uses from the source distribution and between the amount of information exchanged between the boosting algorithm and the weak learner.

Thus, the central question of study in this paper is whether a large dependence on the number of labels $k$ is inherent in the interaction between the weak learner and the boosting algorithm. This question has a direct and significant relationship with the overall complexity of multiclass boosting. Indeed, each call to the weak learner amounts to solving a learning problem over the base class whose complexity is controlled by the prescribed excess loss. Thus, when the number of labels is large, it would be highly beneficial to improve the dependence exhibited by our Algorithm 1.

**An Impossibility Result.**   Interestingly (and somewhat disappointingly), we show that the discrepancy between the sample complexities of the booster and weak learner is inevitable: in Section 5 we prove that, assuming typical limits on the number of weak-learner calls, the number of samples required by the weak learner is at least polynomial in $k$, which is exponentially more than the number of samples needed by the booster. Alternatively, we prove that the weak learner's accuracy parameter must be smaller than an inverse polynomial in $k$, meaning that the returned weak hypotheses must be nearly the best in the base class when $k$ is large. This lower bound follows from a more general trade-off that we prove between the number of oracle calls (to the weak learner) and the sample complexity of the weak learner, meaning that the fewer calls to the weak learner the more that is demanded on each call. Specifically, we show that at least one of these quantities must be $\Omega(\sqrt{k})$.

From a technical perspective, the main challenge in proving this trade-off is the construction of the base class; specifically, the requirement that the base class is easy-to-learn rules out naive randomized constructions ( e.g. as in the oracle-complexity lower bound in [16]), and requires a subtler treatment.

Summarizing, for the model of multiclass boosting that we study, we prove that there is a striking discrepancy between the amount of information (in the form of labeled examples) the boosting algorithm requires from the source distribution and the amount of information exchanged in the channel between the weak learner and the booster.

## 1.2   Related work

The boosting methodology is a fairly mature method, originally formulated for binary classification (e.g. AdaBoost and similar variants) [16]. In contrast with binary classification, there are several versions of multiclass boosting, since there is more than one way to extend the weak learnability assumption. The earlier extensions of boosting to the multiclass setting include AdaBoost.MH and AdaBoost.MR [17], as well as [2]. These works often reduce the $k$-class problem into a single binary problem with a $k$-fold augmented dataset, or to multiple one-versus-all binary problems using Error-Correcting Output Codes (ECOC). The binary reduction can have various problems, including increased complexity, and lack of guarantees of an optimal joint predictor.

When requiring weak learners with classification error less than $1/2$, a reduction to binary problems is unnecessary. Then, AdaBoost.M1 is a straightforward extension of its binary counterpart. We refer the reader to the book by [16] (Chapter 10) for a through survey of the various extensions of AdaBoost to the multiclass setting.

More recent methods [20, 12, 11, 4, 5, 15] were designed from a practical perspective, and provide empirical results of their approaches. Thus, while these works achieve improved performance across various applications, they do not provide a comprehensive theoretical framework for the multiclass boosting problem, and are often based on earlier formulations (e.g., a one-versus-all reduction to the binary setting, or multi-dimensional predictors and codewords).

Notably, previous work by [13] established a theoretical framework for multiclass boosting, which generalizes previous learning conditions. Moreover, as previous boosting methods were shown to require weak learning conditions that are either too strong or too weak, [13] outline an appropriate weak learning condition that is both necessary and sufficient for multiclass boosting. However, this requires the assumption that the weak learner minimizes a tailored loss function that is significantly different from simple classification error. See the next section for a detailed discussion.

## 2   Background

We study boosting algorithms that aim to learn a combination of hypotheses from a fixed base class $\mathcal{H} \subseteq \mathcal{Y}^{\mathcal{X}}$. The data domain $\mathcal{X}$ is finite or countable, and $\mathcal{Y}$ is the finite set of labels $[k] := \{1, ..., k\}$, for $k \geq 2$. In particular, we consider boosting as the task of learning a target concept that can be presented as a *plurality-vote* of hypotheses from $\mathcal{H}$ with margin bounded away from 0. A plurality-vote over the hypotheses class is characterized by a distribution $\lambda$ over $\mathcal{H}$, and the corresponding classifier $\bar{h} = \bar{h}(\lambda)$ is defined by

$$\bar{h}(x) := \underset{\ell \in [k]}{\arg\max} \, \mathbb{P}_{h \sim \lambda}[h(x) = \ell]. \tag{1}$$

The boosting algorithm is given oracle access to $\mathcal{W}$, a weak learner that is an agnostic PAC learner for $\mathcal{H}$. The weakness of $\mathcal{W}$ is manifested in the simplicity of $\mathcal{H}$.[1] Concretely, $\mathcal{W}$ is defined as follows; for every distribution $\mathcal{D}$ over $\mathcal{X} \times \mathcal{Y}$, and $\epsilon^w, \delta^w > 0$, when given a sample of $m^w(\epsilon^w, \delta^w)$ examples drawn i.i.d from $\mathcal{D}$, $\mathcal{W}$ outputs $h \in \mathcal{H}$ such that with probability $1 - \delta^w$,

$$\mathbb{P}_{(x,y) \sim \mathcal{D}}[h(x) = y] \geq \sup_{h^* \in \mathcal{H}} \mathbb{P}_{(x,y) \sim \mathcal{D}}[h^*(x) = y] - \epsilon^w. \tag{2}$$

The weak learner $\mathcal{W}$ considered in this work is defined with respect to the standard classification loss, rather than more complex formulations given in previous work on multiclass boosting. Next, we discuss an alternative weak learning condition that was shown to be necessary and sufficient for multiclass boosting [13].

### 2.1   Optimal weak learnability

We briefly review the formulation introduced by [13] which considers a weak learning condition with respect to a particular gain function, rather than the standard classification objective. Albeit it requires solving a seemingly unnatural classification sub-problem, it was shown to be, in a certain sense, optimal. Consider the following formulation. For any hypothesis $h \in \mathcal{H}$, any labeled example $(x, y) \in \mathcal{X} \times \mathcal{Y}$, and any incorrect label $\ell \in \mathcal{Y}$, define:

$$\sigma_h(x, y, \ell) = \mathbb{1}[h(x) = y] - \mathbb{1}[h(x) = \ell] = \begin{cases} +1, & \text{if } h(x) = y, \\ -1, & \text{if } h(x) = \ell, \\ 0, & \text{otherwise.} \end{cases} \tag{3}$$

Consider the following dual perspective shown by [13], that is associated with the above gain function:

$\gamma$**-margin.**   The $\gamma$*-margin* assumption with respect to a sample $S$, holds if there exists a distribution $\lambda$ over $\mathcal{H}$ (corresponding to a plurality-vote), that has a large margin for every tuple $(x, y, \ell)$, of a labeled example from $S$, and an incorrect label. Concretely,

$$\exists \lambda \in \Delta_{\mathcal{H}}, \quad \forall (x, y) \in S, \ell \neq y, \qquad \mathbb{E}_{h \sim \lambda}\big[\sigma_h(x, y, \ell)\big] \geq \gamma. \tag{4}$$

$\gamma$**-edge of $\mathcal{H}$.**   The edge is the gap between the performance of a certain hypothesis over random prediction. Define $\tilde{S} = \{(x, \ell) | (x, y) \in S, \ell \neq y\}$ the set of all incorrectly-labeled examples (with respect to $S$). The *empirical $\gamma$-weak learnability* assumption of $\mathcal{H}$ holds, if for any distribution $\tilde{\mathcal{D}}$ over $\tilde{S}$, there is a hypothesis $h \in \mathcal{H}$ that has an *edge* of $\gamma$ (as is typically referred to in the binary boosting setting[2]), with respect to $\tilde{\mathcal{D}}$, over random prediction. Concretely,

$$\forall \tilde{\mathcal{D}} \in \Delta_{\tilde{S}}, \qquad \exists h \in \mathcal{H}, \qquad \mathbb{E}_{(x,\ell) \sim \tilde{\mathcal{D}}}\big[\sigma_h(x, y, \ell)\big] \geq \gamma. \tag{5}$$

---

[1]The simplicity of the base class is measured via its Natarajan dimension, an extension of the VC-dimension to the non-binary setting. See Section 4 for the formal definition and discussion.

[2]Note that for a uniformly-random predictor $f$, and any distribution $\mathcal{D}$, $\underset{(x,y,\ell) \sim \mathcal{D}}{\mathbb{E}}[\sigma_f(x, y, \ell)] = 0$.

Previous work by [13] has shown via a minimax argument that, for a given training set $S$, for the realizability of $S$ by some $\gamma$-margin plurality-vote $\bar{h}$ to hold, i.e., for equation (4) to hold, it is both a sufficient and a necessary condition that equation (5) is satisfied. That is, (4) holds if and only if (5) holds. Note that these observations are only concerning a fixed training set $S$. Next, we extend these ideas using the following notion of a $\gamma$-realizable distribution.

**Definition 1** ($\gamma$-*realizable sample/distribution*). *Let $\gamma \in (0,1)$, $\mathcal{H} \subseteq \mathcal{Y}^\mathcal{X}$ a hypotheses class. A sample $S \subseteq \mathcal{X} \times \mathcal{Y}$ is $\gamma$-realizable with respect to $\mathcal{H}$ if Equation (4) holds. A distribution $\mathbf{D}$ over $\mathcal{X} \times \mathcal{Y}$ is $\gamma$-realizable with respect to $\mathcal{H}$ if any independent sample from it is $\gamma$-realizable.*[3]

The end goal of the boosting algorithm is to approximate a target $\gamma$-realizable distribution $\mathbf{D}$ with respect to classification loss. The booster is provided with oracle access to a learner for the base class $\mathcal{H}$, as well as $\epsilon^B, \delta^B > 0$ and $m(\epsilon^B, \delta^B)$ examples drawn i.i.d. from $\mathcal{D}$. The booster makes $T$ iterative oracle calls to the learner and outputs a plurality-vote $\bar{h}_T$ such that with probability $1 - \delta^B$,

$$\mathbb{P}_{(x,y)\sim\mathcal{D}}[\bar{h}_T(x) \neq y] \leq \epsilon^B. \tag{6}$$

## 3 Multiclass Adaboost: algorithm and analysis

We give an efficient adaptive algorithm for multiclass boosting, which converts an agnostic-PAC learner for the base class $\mathcal{H}$, to a learner for any target distribution that is $\gamma$-realizable with respect to $\mathcal{H}$. Towards that end, we first give a reduction to the framework provided by [13].

Specifically, we show the that the $\sigma$-gain weak-learning condition (defined in Equation (5)), can be captured by a standard classification loss, agnostic PAC learner for $\mathcal{H}$ (Equation (2)).

**Lemma 2.** *Let $S \subseteq \mathcal{X} \times \mathcal{Y}$ be a training set, and $\tilde{S} = \{(x, \ell) | (x, y) \in S, \ell \neq y\}$ the corresponding set of incorrectly-labelled examples. Let $\mathcal{W}$ be an agnostic-PAC learner for $\mathcal{H} \subseteq \mathcal{Y}^\mathcal{X}$. Let $\epsilon^w, \delta^w > 0$, and edge $\gamma > 0$, and assume that $S$ is a $\gamma$-realizable sample w.r.t. $\mathcal{H}$. Then, for any distribution $\tilde{\mathcal{D}}$ over $\tilde{S}$, there is a distribution $\mathcal{D}$ over $S \cup \tilde{S}$, such that when given a sample of $m_w \geq m_w(\varepsilon^w, \delta^w)$ examples drawn i.i.d from $\mathcal{D}$, $\mathcal{W}$ outputs $h \in \mathcal{H}$ s.t. with probability $1 - \delta^w$,*

$$\mathbb{E}_{(i,\ell)\sim\tilde{\mathcal{D}}}\big[\sigma_h(x_i, y_i, \ell)\big] \geq \gamma - \varepsilon^w k.$$

Observe that our conversion of a classification loss learner to a $\sigma$-gain learner comes at a cost, as it incurs a multiplicative overhead of factor $k$ in the excess error. The implications of that fact on the sample complexity of the learner are further described in Section 4.

Next, we present our multiclass boosting approach, described in Algorithm 1. We remark that AdaBoost.MR [17] a previous boosting method that is a version of AdaBoost for ranking loss, applies a similar technique as described in Algorithm 1. However, their algorithm requires access to a variant of the $\sigma$-gain learner for $\mathcal{H}$, whereas our method only assumes a standard classification loss learner.

In the following result, formally stated in Theorem 3, we show that the predictor outputted by the boosting algorithm, given in Algorithm 1, is consistent with the training data.

**Theorem 3.** *(Upper bound - Consistency) Let $\mathcal{W}$ be an agnostic-PAC learner for the base class $\mathcal{H} \subseteq \mathcal{Y}^\mathcal{X}$. Let $\gamma, \delta > 0$, and training set $S$ of $m$ labelled examples. If $S$ is $\gamma$-realizable with respect to $\mathcal{H}$ (see Definition 1), then applying Algorithm 1 with $T \geq \frac{8(\log(m)+\log(k))}{\gamma^2}$, and $m_w = m_w(\gamma/k, \delta/T)$, outputs with probability at least $1 - \delta$, an $\bar{h}_T$ that is consistent with $S$.*

---

[3]Our results remain valid for a relaxed definition of a $\gamma$-realizable distribution for which a $\gamma$-realizable samples are only drawn with high probability. However, we consider the above for simplicity of presentation.

---

**Algorithm 1** Multiclass Adaboost

---

1: Input: $S = \{(x_1, y_1), \ldots, (x_m, y_m)\}$.
2: Initialize: $\tilde{\mathcal{D}}_1(i, \ell) = \frac{1}{m(k-1)}$ if $\ell \neq y_i$, else set 0, for $\ell \in [k], i \in [m]$.
3: **for** $t = 1, \ldots, T$ **do**
4:    Obtain $\mathcal{D}_t$ from $\tilde{\mathcal{D}}_t$ (see Lemma 2), and pass $m_w$ examples to $\mathcal{W}$ drawn i.i.d from $\mathcal{D}_t$.
5:    Let $h_t$ be the weak hypothesis returned by $\mathcal{W}$, compute its empirical edge:

$$\gamma_t = \mathbb{E}_{(i,\ell) \sim \tilde{\mathcal{D}}_t} \big[ \sigma_{h_t}(x_i, y_i, \ell) \big].$$

6:    Set $\alpha_t = \frac{1}{2} \ln \left( \frac{1+\gamma_t}{1-\gamma_t} \right)$. Update, for all $i \in [m], y_i \neq \ell \in [k]$:

$$\tilde{\mathcal{D}}_{t+1}(i, \ell) = \frac{\tilde{\mathcal{D}}_t(i, \ell)}{Z_t} \times \begin{cases} e^{-\alpha_t}, & \text{if } h_t(x_i) = y_i, \\ e^{\alpha_t}, & \text{if } h_t(x_i) = \ell, \\ 1, & \text{otherwise.} \end{cases}$$

$$= \frac{\tilde{\mathcal{D}}_t(i, \ell) \exp(-\alpha_t \cdot \sigma_{h_t}(x_i, y_i, \ell))}{Z_t},$$

where $Z_t$ is a normalization factor (chosen so that $\tilde{\mathcal{D}}_{t+1}$ will be a distribution).
7: **end for**
8: Output the weighted plurality-vote,

$$\bar{h}_T(x) := \operatorname*{argmax}_{\ell \in [k]} \left( \sum_{t=1}^{T} \alpha_t \cdot \mathbb{1}[h_t(x) = \ell] \right).$$

---

**Runtime analysis.** The number of iterations required for Algorithm 1 to succeed has only a logarithmic dependence on the number of labels $k$. Nonetheless, since the algorithm maintains a distribution $\tilde{\mathcal{D}}$ over the data, of size $m \times k$, it might seem as if the cost of each round is linearly dependent on $k$. However, observe that the updates to distribution $\tilde{\mathcal{D}}$ only pertain to incorrect labels $\ell$ which at least one of $h_1, \ldots, h_t$ mistakenly predicted to be correct. Thus, at most $mT$ such values are being updated. Moreover, note that performing a random sample from $\mathcal{D}_t$ (step 4 of Algorithm 1) can be done in time $O(\log(mk))$ [7], as $\tilde{\mathcal{D}}, \mathcal{D}$ maintained by the algorithm can be kept sorted throughout the iterations. Therefore, the running time of each round $t$ is $O(m \log(mk))$, and the overall running time only has poly-logarithmic dependence on $k$.

## 4  Sample complexity upper bounds

In the previous section we have focused on the consistency model, and have shown that a multiclass boosting algorithm with a running time of only log dependence on $k$, can achieve zero training error. In this section we examine the generalization aspects of the algorithm. The results described below have two main implications, concerning the sample-complexity required for the boosting algorithm as a black-box (subsection 4.2), and the sample-complexity of the weak-learner required in each iteration, within the boosting framework (subsection 4.1).

We show that on the one hand, the sample complexity required for the booster defined in Algorithm 1 to learn a $\gamma$-realizable distribution $\mathbf{D}$ over $\mathcal{X} \times \mathcal{Y}$, is upper bounded by $m = O(\log(k))$. However, the upper bound we give on sample complexity of the weak learning oracle $\mathcal{W}$ is $m_w = O(k^2)$, omitting other terms. This striking contrast naturally raises the question of whether a different boosting approach exists, such that the large dependence on $k$ is significantly improved to match the logarithmic upper bound of the algorithm. In Section 5 we provide a lower bound showing that these gaps are inherent to the problem of multiclass boosting.

**Natarajan dimension of $\mathcal{H}$.** In the binary case, the VC-dimension characterizes the sample complexity of hypothesis classes. In the multiclass case, [14] introduced an extension of the VC-dimension to non-binary functions, defined as follows.

**Definition 4.** *Let $\mathcal{H} \subseteq \mathcal{Y}^{\mathcal{X}}$ a hypothesis class and let $S \subseteq \mathcal{X}$. We say that $\mathcal{H}$ N-shatters $S$ if $\exists \pi_1, \pi_2 : S \to \mathcal{Y}$ such that $\forall x \in S$, $\pi_1(x) \neq \pi_2(x)$, and for every $T \subseteq S$ there is $h \in \mathcal{H}$ such that,*

$$\forall x \in T, \ h(x) = \pi_1(x), \ and \ \forall x \in S \setminus T, \ h(x) = \pi_2(x).$$

*The Natarajan dimension $d_N(\mathcal{H})$, is the maximal cardinality of a set that is N-shattered by $\mathcal{H}$.*

The Natarajan dimension coincides with the VC-dimension for $k = 2$. The Natarajan dimension of a hypothesis class, $d_N(\mathcal{H})$, characterize the sample complexity required for PAC-learning the class $\mathcal{H}$, up to a multiplicative factor of $O(\log(k))$. Specifically, by [6, 9] the sample complexity required to agnostically-PAC learn a class $\mathcal{H}$ is upper bounded by,

$$m_{\mathcal{H}}(\epsilon, \delta) = O\left(\frac{d_N(\mathcal{H})\log(k) + \ln(1/\delta)}{\epsilon^2}\right). \tag{7}$$

### 4.1 The sample complexity of weak-learning

In section 3 we have discussed a reduction of a classification-loss learner to a $\sigma$-gain learner. However, this transformation required a multiplicative overhead of factor $k$ in the excess error. Observe that by plugging in any $\epsilon^w < \gamma/k$, we get that the hypothesis returned by $\mathcal{W}$ has a strictly positive edge, which allows boosting. The implications of that fact on the sample complexity of $\mathcal{W}$ are summarized in Remark 5, which immediately follows from Theorem 3, Lemma 2, and Equation (7).

**Remark 5.** *Given oracle access to an agnostic-PAC learner $\mathcal{W}$ for $\mathcal{H}$, and sample set $S$ of size $m$, applying Algorithm 1 with $T = O(\log(mk)/\gamma^2)$ oracle calls to $\mathcal{W}$, setting $\epsilon_t^w < \gamma/k$, and passing $m_w = O\left(\frac{d_N(\mathcal{H})\log(k)}{\gamma^2} \cdot k^2\right)$ examples to $\mathcal{W}$, yields an output $\bar{h}_T$ is consistent with $S$.*

### 4.2 The sample complexity of Boosting

Next, we show that the sample complexity required for multiclass boosting to learn a $\gamma$-realizable distribution $\mathbf{D}$ over $\mathcal{X} \times \mathcal{Y}$, does only require a logarithmic dependence on $k$. This bound on the sample complexity is given by considering the output predictor in Algorithm 1 and bound the sample complexity that is required for it to generalize.

In the following result, formally stated in Theorem 6, we bound the generalization error of a predictor outputted by the boosting algorithm, given in Algorithm 1. In particular, we use a margins-based analysis of the generalization error (see margins-based analysis for the binary case in [16], chapter 5.2), utilizing two key facts; (i) the predictor $\bar{h}_T$ obtained by Algorithm 1 has a large empirical margin, and (ii) uniform convergence applies to the margin of $\bar{h}_T$. Moreover, the large large margin property both holds empirically, and generalizes, while requiring only a logarithmic dependence on $k$. These facts are formally stated and proved in the appendix.

**Theorem 6.** *(Upper bound - Generalization) Let $\mathbf{D}$ be a distribution over $\mathcal{X} \times \mathcal{Y}$ that is $\gamma$-realizable by a hypotheses class $\mathcal{H} \subseteq \mathcal{Y}^{\mathcal{X}}$, and $\gamma > 0$. Let $\epsilon, \delta > 0$, and the training set $S$ be a random sample of $m = \Theta\left(\frac{d_N(\mathcal{H})\log(k)\log(\frac{1}{\epsilon})}{\gamma^2 \epsilon^2} + \frac{\log(1/\delta)}{\epsilon^2}\right)$ examples drawn i.i.d from $\mathbf{D}$. Then, with probability $1 - \delta$, over the random choice of the training set $S$, applying Algorithm 1 with $T = \Theta(\frac{\log(mk)}{\gamma^2})$, outputs a predictor $\bar{h}_T$ such that,*

$$\mathbb{P}_{(x,y) \sim \mathbf{D}}\left[\bar{h}_T(x) \neq y\right] \leq \epsilon.$$

Note that although Theorem 6 pertains to the complexity required for learning via a *boosting* algorithm; a similar statement can even be verified to hold for any ERM learner for $\mathcal{H} \subseteq \mathcal{Y}^{\mathcal{X}}$, which can learn any $\gamma$-realizable for $\mathcal{H}$, with a similar sample complexity. This follows from the definition of $\gamma$-realizability, the fact that a sparse plurality-vote can be sampled from the realizable one (a proof technique also used in Theorem 6), and the sample complexity bound given (7). Overall, we get that the task of multiclass learning, either directly or via boosting, has a logarithmic dependence on $k$.

## 5 Matching Lower Bounds

In this section, we give lower bounds on the sample and oracle complexities required for a boosting algorithm to attain a small generalization error, for a large number of labels $|\mathcal{Y}| = k$. Alternatively,

such bounds can be inverted to give a a lower bound on the generalization error that can be achieved by a boosting algorithm for a fixed sample complexity and running time. Thus, these bounds characterize the efficiency of any multiclass boosting algorithm, in terms of how the number of rounds and samples must depend on the desired accuracy, with respect to $k$.

## 5.1 Model definition

Recall that in previous sections, a weak learning algorithm $\mathcal{W}$ was defined to be an agnostic-PAC learner for a hypothesis class $\mathcal{H}$, which satisfies Equation (2). Towards the goal of providing a lower bound for a general boosting algorithm, we consider two alternative, standard models of interaction between a booster and weak learner, that only differ in the input the weak learner expects to receive.

1. **Boosting-by-resampling** - in this form of interaction, the oracle $\mathcal{W}$ is a learning algorithm for $\mathcal{H} \subseteq \mathcal{Y}^{\mathcal{X}}$ such that when given a set of $m_w(\epsilon_w, \delta_w = 1/3)$ labelled examples drawn i.i.d. from a distribution $\mathcal{D}$ over $\mathcal{X} \times \mathcal{Y}$, it returns an $\epsilon_w$-optimal hypothesis in $\mathcal{H}$, with respect to the classification loss over $\mathcal{D}$. In this setting, we consider the learner $\mathcal{W}$ to simply be an ERM (empirical risk minimizer) for a given sample.

2. **Boosting-by-reweighting** - in this model, the learner $\mathcal{W}$ is allowed to directly utilize the distribution over examples $\mathcal{D}$, so that the weighted training error is approximately minimized explicitly, but $\mathcal{W}$ may still return any $\epsilon_w$-optimal hypothesis in $\mathcal{H}$ over $\mathcal{D}$.

In addition, we now define what is a general boosting algorithm here, for either interaction form. A boosting algorithm $B$ is one which, when provided with oracle access a weak learner $\mathcal{W}$ for $\mathcal{H}$ (in either model of interaction; resampling or reweighting), as well as $\epsilon^B, \delta^B > 0$, and a training set $S$ of $m = m(\epsilon^B, \delta^B)$ examples drawn i.i.d. from a target distribution $\mathbf{D}$ over $\mathcal{X} \times \mathcal{Y}$ that is $\gamma$-realizable (see Definition 1), will with probability at least $1 - \delta^B$, output a combined hypothesis that is a plurality-vote of hypotheses $h \in \mathcal{H}$ obtained via the learner $\mathcal{W}$, such that its classification error with respect to $\mathbf{D}$ is at most $\epsilon^B$. In our construction, we simply fix $\epsilon^B = 2\gamma$, and $\delta^B = 1/2$.

Although the boosting algorithm may be randomized, we can regard its randomization source, denoted $\mathcal{R}$, as itself an input to the algorithm. Thus, we view $B$ as a fixed and deterministic function of the training set $S$, the internal randomization $\mathcal{R}$, and the responses obtained via the learning oracle $\mathcal{W}$.

Note that we assume $B$ does not know the hypothesis class $\mathcal{H}$, and its only access to it is through its access to the learning oracle $\mathcal{W}$. However, it is allowed to have other information about the weak learner, such as $m^w$ the required sample size of $\mathcal{W}$, the associated edge $\gamma$, etc. Note that we also restricted the booster such that it may only output a plurality-vote of hypotheses $h \in \mathcal{H}$ obtained via the learner $\mathcal{W}$. Aside from the above, the boosting algorithm $B$ is entirely unrestricted.

## 5.2 Results

We derive lower bounds using a simple construction of a class $\mathcal{H}$, and a base learner $\mathcal{W}$, applied in both models described above; by-resampling and by-reweighting. The main idea of the construction is of designing a simple class $\mathcal{H}$, such that its hypotheses will make diverse mistakes which the booster cannot know in advance. This forces the booster to pick a distribution $\mathcal{D}_t$ with a large support of $\Omega(\sqrt{k})$ elements, and require high accuracy $\epsilon_t \approx 1/k$. Otherwise, it can be shown no matter how the obtained hypotheses are combined in a plurality-vote, there will always be a large mistake, unless a large number of rounds $T = \Omega(\sqrt{k})$ is used. This trade-off between oracle and sample complexities provides the lower bound we seek, showing that any booster incurs a cost of $\Omega(\sqrt{k})$.

**Theorem 7.** *(**Lower bound**) Let $\gamma \in (0, 1/2)$ denote the margin parameter, $k > 2$ number of labels, and assume that $k > 2/\gamma$. Let $T > 0$ denote the number of iterations. For any boosting algorithm $B$, by-reweighting or by-resampling, there exists a class $\mathcal{H} \subseteq \mathcal{Y}^{\mathcal{X}}$, an agnostic-PAC learner $\mathcal{W}$ for $\mathcal{H}$, and a target $\frac{\gamma}{2}$-realizable distribution $\mathcal{D}$, such that; If $B$ is applied with oracle access to $\mathcal{W}$ for $T$ iterations, and for all $t \leq T$, $B$ picks error rates $\epsilon_t^w > \theta := \frac{2T}{k-1/\gamma-1}$ (reweighting model), or picks sample size $m_t^w < 1/\theta$ (resampling model), and then outputs a final hypothesis $\bar{h}_B$, then,*

$$\mathbb{P}_{S,\mathcal{R}} \left[ \mathbb{P}_{\mathcal{D}} \left[ \bar{h}_B(x) \neq y \right] > 2\gamma \right] \geq 1/2,$$

*where $S$ is the initial random training sample drawn from $\mathcal{D}$, and $\mathcal{R}$ denotes the internal randomization of the boosting algorithm.*

The trade-off between the oracle and sample complexity bounds is demonstrated in Corollary 8 next.

**Corollary 8.** *(**Lower bound - complexity trade-off**) By applying Theorem 7 with different values of $T$, we get that for any boosting algorithm $B$ to achieve error rate at most $2\gamma$,*

1. *There exists a class $\mathcal{H}$ and learner $\mathcal{W}$, s.t. if $B$ makes $T = O(\frac{1}{\gamma})$ oracle calls to $\mathcal{W}$, then there is some $t \le T$ in which $\mathcal{W}$ must incur a sample complexity of $m_t^w = \Omega(k)$.*

2. *There exists a class $\mathcal{H}$ and learner $\mathcal{W}$, s.t. either $B$ makes $T = \Omega(\sqrt{k})$ oracle calls to $\mathcal{W}$, or there is some $t \le T$ in which $\mathcal{W}$ must incur a sample complexity of $m_t^w = \Omega(\sqrt{k})$.*

3. *There exists a class $\mathcal{H}$ and learner $\mathcal{W}$, s.t. if for all $t \le T$, the sample complexity of $\mathcal{W}$ is $m_t^w = O(1)$, then $B$ makes $T = \Omega(\sqrt{k})$ oracle calls to $\mathcal{W}$.*

Recall that $k$ is thought of as the very large, dominating term, compared to $1/\gamma$. However, the results shown in Corollary 8 only rely on the assumption $k > 1/\gamma$. We remark that if $k \gg 1/\gamma$, the interplay between the time and sample complexity bounds changes accordingly. For example, if $k > 1/\gamma^2$, it can be similarly shown that the first item of Corollary 8 holds for $T = O(\frac{1}{\gamma^2})$ as well.

## 5.3 Proof overview

Below is an informal, high-level, overview of the proof. We refer the reader to the appendix for the formal statements and proofs. We begin with describing the construction of the class $\mathcal{H}$. Let $\mathcal{X} = \{x_1, \ldots, x_n\}$ be the domain, with $n = \frac{1}{2\gamma}$ data points, and assume that $n \ll k$. We fix the constant function $f(x) = 1$ to be the target labelling. Then, the labelled data is $\{(x_1, 1)\ldots(x_n, 1)\}$, and let the target distribution $\mathcal{D}$ be the uniform distribution over it. We note that our construction and the lower bound argument hold even for boosting algorithms that have full knowledge of the data and distribution $\mathcal{D}$. However, recall that the boosting algorithm is restricted to only output a weighted plurality-vote of hypotheses $h^1, \ldots h^T \in \mathcal{H}$ obtained via $\mathcal{W}$. Next, consider hypotheses of the following form; for all $j \in [n]$ and $\ell \in \{2, \ldots, k\}$, define

$$h_{j,\ell}(x) = \begin{cases} 1 & \text{if } x = x_j, \\ \ell & \text{otherwise.} \end{cases} \tag{8}$$

We denote the set of all such hypotheses as $\mathcal{H}_{all}$. The base-class $\mathcal{H}$ will be a carefully chosen random subset of $\mathcal{H}_{all}$. The key properties that guide the construction of $\mathcal{H}$ are: (i) a uniform plurality vote over all hypotheses in $\mathcal{H}$ yields the target-function $f$ with margin $\Omega(\gamma)$, and (ii) every weighted plurality vote of a proper subset of $\mathcal{H}$ is far-away from $f$.

We then use the probabilistic method to randomly construct $\mathcal{H}$. Specifically, the class $\mathcal{H}$ contains $2n - 2$ carefully chosen hypotheses that are fixed, and a single random hypothesis, $h_{j,L}$, where $L$ is drawn uniformly at random from the set of labels $2n, \ldots, k$ (recall that $k \gg n$). We stress that the $2n - 2$ hypotheses, and the index $j$, are chosen such that for each possible $L$, the obtained class satisfies the two properties noted above.

Observe that the class $\mathcal{H}_{all}$ is an easy-to-learn class, and it can be verified that it has a constant Natarajan dimension $= 2$, and that it is learnable by any ERM with sample complexity independent of $k$. Since $\mathcal{H} \subset \mathcal{H}_{all}$ for any $L$, it exhibits theses traits as well. Moreover, by property (i) discussed above, it holds that there exists a plurality-vote over $\mathcal{H}$ that is consistent with the data, with margin $\gamma/2$. That is, the distribution $\mathcal{D}$ which is uniform over $\mathcal{X} \times \{1\}$ is $\gamma/2$-realizable, with respect to $\mathcal{H}$.

We refer to the randomly chosen hypothesis $h_{j,L}$ as the *hidden* hypothesis. Note that if the booster cannot "discover" the hidden hypothesis throughout its run (i.e., obtain it via the weak learner), then by property (ii) of $\mathcal{H}$, any weighted plurality it can output must have large error. Thus, it suffices to exhibit an agnostic PAC learner for $\mathcal{H}$, for which the booster will fail to obtain the hidden hypothesis unless the accuracy/sample-complexity are high.

**The "finder-chooser" game.** The above construction forces any successful boosting algorithm to find the hidden hypothesis. This is the crucial observation needed for our proof and it enables us to

design a weak learner which reduces the task of finding the hidden hypothesis to the following simple "finder-chooser" game. This is an iterative game in which a *"chooser"* player picks a "hidden" element $L$ out of a set of elements $K$, of size $|K| = k$, and a *"finder"* player iteratively attempts to discover it. Specifically, the finder is only required to find any small subset of elements from $K$ that the hidden element $L$ belongs to. Towards that end, in each round the finder queries the chooser, and the chooser replies with a binary response, which indicates if the chooser has indeed picked the hidden element. We consider two different query-models in which the finder is allowed to interact with the chooser; via weights, or via examples (which correspond to the alternative model of boosting, discussed in subsection 5.1). In both cases, the finder is given a binary response in each round, of whether or not the hidden element is in the subset it picked. If the finder failed, it continues to the next round, otherwise it succeeded to find a subset which contains $L$, and the game ends.

It is clear that there is a trade-off between the size of the picked subsets and the number of rounds needed to find the containing subset. We formalize this trade-off, and give bounds on the probability that the finder succeeds, when the chooser is uniform over $K$. This in turn, yields the desired bounds by reduction to the boosting setting.

# 6 Conclusion

We study the problem of multiclass boosting, where we focus on the case where the number of labels $k \gg 2$ is large, and study how $k$ affects complexity aspects of boosting. Our first result is a natural extension of Adaboost to the multiclass case, using a simple weak learning oracle. We then show that there is a prominent gap between the sample complexities of the weak learner $O(k^2)$, and of the boosting algorithm $O(\log(k))$, that are needed for learning. Finally, we give a lower bound which shows that indeed, even for a trivial (constant-time) learning tasks of both the base class and the target, learning via boosting is a more challenging task. That is, a large dependence on $k$ is in fact an inherent cost required for multiclass boosting.

## Acknowledgments and Disclosure of Funding

S. Moran is a Robert J. Shillman Fellow and is supported by the ISF, grant no. 1225/20, by an Azrieli Faculty Fellowship, and by BSF grant 2018385. Part of this work was done when S. Moran was at Google AI Princeton. N. Brukhim and E. Hazan acknowledge the support of NSF grant 1704860.

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
