# A Omitted proofs of Section 3

## A.1 Proof of Lemma 2

*Proof.* Assume $|S| = m$, for some $m > 0$. Then, corresponding set of incorrectly-labelled examples is of size $|\tilde{S}| = m(k-1)$. Let $\tilde{\mathcal{D}} \in \Delta_{\tilde{S}}$, be any distribution over $m \times (k-1)$. We show that we can construct a distribution $\mathcal{D} \in \Delta_{\tilde{S} \cup S}$ over $m \times k$, for which the guarantee in the Lemma holds.

First, observe that there must exist $a \in \Delta_m$ and $b_1, ..., b_m \in \Delta_{(k-1)}$, such that for all $i, j$, we have $\tilde{\mathcal{D}}(i, j) = a(i)b_i(j)$. For every $i \in [m]$, let $b_i' \in \Delta_k$ be a distribution constructed as follows:

$$b_i'(j) = \begin{cases} \frac{2}{k} & \text{if } j = y_i \\ \frac{1}{k} - \frac{b_i(j)}{k} & \text{otherwise} \end{cases} \tag{9}$$

Then, we set $\mathcal{D}$ such that for all $i, j$, we have $\mathcal{D}(i, j) := a(i)b_i'(j)$. Then, observe that for any $h \in \mathcal{H}$,

$$\mathbb{P}_{\mathcal{D}}[h(x) = y] = \sum_{i=1}^{m} \sum_{j=1}^{k} a(i)b_i'(j)\mathbb{1}[h(x_i) = j] \qquad \text{(By substituting } \mathcal{D} \text{ for } a, b')$$

$$= \sum_{i=1}^{m} a(i) \left( \frac{2}{k}\mathbb{1}[h(x_i) = y_i] + \sum_{j \neq y_i} \frac{1 - b_i(j)}{k}\mathbb{1}[h(x_i) = j] \right) \qquad \text{(By (9))}$$

$$= \frac{2}{k} \mathbb{P}_{\tilde{\mathcal{D}}}[h(x) = y] + \frac{1}{k}(1 - \mathbb{P}_{\tilde{\mathcal{D}}}[h(x) = y]) - \frac{1}{k} \sum_{i=1}^{m} \sum_{j \neq y_i} \tilde{\mathcal{D}}(i, j)\mathbb{1}[h(x_i) = j] \tag{10}$$

$$= \frac{1}{k} \mathbb{P}_{\tilde{\mathcal{D}}}[h(x) = y] + \frac{1}{k} - \frac{1}{k} \sum_{i=1}^{m} \sum_{j \neq y_i} \tilde{\mathcal{D}}(i, j)\mathbb{1}[h(x_i) = j] \tag{11}$$

$$= \frac{1}{k} + \frac{1}{k} \left( \mathbb{P}_{\tilde{\mathcal{D}}}[h(x) = y] - \sum_{i=1}^{m} \sum_{j \neq y_i} \tilde{\mathcal{D}}(i, j)\mathbb{1}[h(x_i) = j] \right) \tag{12}$$

$$= \frac{1}{k}(1 + \mathbb{E}_{(i,j) \sim \tilde{\mathcal{D}}}[\sigma_h(x_i, y_i, j)]). \tag{13}$$

By running the learner $\mathcal{W}$ over $m^w$ i.i.d. examples generated by $\mathcal{D}$, the algorithm returns a hypothesis $h \in \mathcal{H}$ such that, with probability of at least $1 - \delta^w$ (over the sampling of the $m^w$ examples from $\mathcal{D}$), we have,

$$\mathbb{P}_{\mathcal{D}}[h(x) = y] \geq \max_{h^* \in \mathcal{H}} \mathbb{P}_{\mathcal{D}}[h^*(x) = y] - \epsilon^w. \tag{14}$$

Then, by applying the transformation given in (13) to (14) on both size, and re-arrange terms, we get,

$$\mathbb{E}_{(i,\ell) \sim \tilde{\mathcal{D}}}[\sigma_h(x_i, y_i, \ell)] \geq \max_{h^* \in \mathcal{H}} \mathbb{E}_{(i,\ell) \sim \tilde{\mathcal{D}}}[\sigma_{h^*}(x_i, y_i, \ell)] - \epsilon^w k \geq \gamma - \epsilon^w k,$$

where the last inequality follows by the assumption that $S$ is $\gamma$-realizable with respect to $\mathcal{H}$ (see Definition 1). $\square$

## A.2 Proof of Theorem 3

*Proof.* For any $x \in \mathcal{X}, \ell \in [k]$, we define:

$$F(x, \ell) = \sum_{t=1}^{T} \alpha_t \mathbb{1}[h_t(x) = \ell]. \tag{15}$$

Unraveling the recurrence in Algorithm 1 that defines $\tilde{\mathcal{D}}_{t+1}$ in terms of $\tilde{\mathcal{D}}_t$ gives,

$$
\begin{aligned}
\tilde{\mathcal{D}}_{T+1}(i,\ell) &= \tilde{\mathcal{D}}_1(i,\ell) \times \frac{e^{-\alpha_1 \cdot \sigma_{h_1}(x_i,y_i,\ell)}}{Z_1} \times \ldots \times \frac{e^{-\alpha_T \cdot \sigma_{h_T}(x_i,y_i,\ell)}}{Z_T} \\
&= \tilde{\mathcal{D}}_1(i,\ell) \times \frac{e^{-\sum_{t=1}^T \alpha_t \cdot \sigma_{h_t}(x_i,y_i,\ell)}}{\prod_{t=1}^T Z_t} \\
&= \tilde{\mathcal{D}}_1(i,\ell) \times \frac{e^{-\sum_{t=1}^T \alpha_t \cdot \left(\mathbb{1}[h_t(x_i)=y_i] - \mathbb{1}[h_t(x_i)=\ell]\right)}}{\prod_{t=1}^T Z_t} \\
&= \tilde{\mathcal{D}}_1(i,\ell) \times \frac{e^{-F(x_i,y_i)+F(x_i,\ell)}}{\prod_{t=1}^T Z_t}.
\end{aligned}
\tag{16}
$$

Next, note that $\bar{h}_T(x) = \mathrm{argmax}_{\ell \in [k]} F(x,\ell)$. Therefore, for any $i \in [m]$ and $\ell \neq y_i$, we get that if $\bar{h}_T(x_i) = \ell$, then $F(x_i,\ell) \geq F(x_i,y_i)$. This implies that $e^{F(x_i,\ell)-F(x_i,y_i)} \geq 1$. Hence, we have, $\mathbb{1}[\bar{h}_T(x_i) = \ell] \leq e^{F(x_i,\ell)-F(x_i,y_i)}$. We can now bound the label-weighted error:

$$
\sum_{i=1}^m \sum_{\ell \neq y_i} \tilde{\mathcal{D}}_1(i,\ell) \cdot \mathbb{1}[\bar{h}_T(x_i) = \ell] \leq \sum_{i=1}^m \sum_{\ell \neq y_i} \tilde{\mathcal{D}}_1(i,\ell) \cdot e^{F(x_i,\ell)-F(x_i,y_i)}
\tag{17}
$$

$$
= \sum_{i=1}^m \sum_{\ell \neq y_i} \tilde{\mathcal{D}}_{T+1}(i,\ell) \prod_{t=1}^T Z_t
\tag{18}
$$

$$
= \prod_{t=1}^T Z_t,
\tag{19}
$$

where equation (18) uses equation (16), and equation (19) uses the fact that $\tilde{\mathcal{D}}_{T+1}$ is a distribution which sums to 1 (over all examples and their *incorrect* labels). Denote $q_t = \sum_{i=1}^m \sum_{\ell \neq y_i} \tilde{\mathcal{D}}_t(i,\ell) \cdot \mathbb{1}[h_t(x_i) = \ell]$. Then, bounding the normalization factor,

$$
Z_t = \sum_{i=1}^m \sum_{\ell \neq y_i} \tilde{\mathcal{D}}_t(i,\ell) \cdot e^{-\alpha_t \cdot \sigma_{h_t}(x_i,y_i,\ell)}
\tag{20}
$$

$$
= \sum_{i=1}^m \sum_{\ell \neq y_i} \tilde{\mathcal{D}}_t(i,\ell) \cdot \left( \mathbb{1}[h_t(x_i)=y_i] \cdot e^{-\alpha_t} + \mathbb{1}[h_t(x_i)=\ell] \cdot e^{\alpha_t} + \mathbb{1}[h_t(x_i) \notin \{y_i,\ell\}] \right)
\tag{21}
$$

$$
= (q_t + \gamma_t) \cdot e^{-\alpha_t} + q_t \cdot e^{\alpha_t} + (1 - 2q_t - \gamma_t),
\tag{22}
$$

where equation (22) simply follows by the definition of $q_t$ and $\gamma_t$ (step 5 of Algorithm 1). Observe that by plugging our choice of $\alpha_t$ in to equation (22) and re-arranging terms, we get that the coefficient of $q_t$ is: $\frac{2}{\sqrt{1-\gamma_t^2}} - 2$, which is a positive term for any $\gamma_t \in (0,1)$. Therefore, equation (22) is a monotonic increasing function of $q_t$. Moreover, since $(1 - 2q_t - \gamma_t) \geq 0$, we have $q_t \leq \frac{1}{2} - \frac{\gamma_t}{2}$. Thus, we get,

$$
Z_t \leq e^{-\alpha_t} \cdot \left( \frac{1}{2} + \frac{\gamma_t}{2} \right) + e^{\alpha_t} \cdot \left( \frac{1}{2} - \frac{\gamma_t}{2} \right) = \sqrt{1 - \gamma_t^2}.
\tag{23}
$$

Lastly, plugging into equation (19) gives,

$$
\sum_{i=1}^m \sum_{\ell \neq y_i} \tilde{\mathcal{D}}_1(i,\ell) \mathbb{1}[\bar{h}_T(x_i) = \ell] \leq \prod_{t=1}^T \sqrt{1 - \gamma_t^2}
\tag{24}
$$

$$
\leq e^{-1/2 \cdot \sum_{t=1}^T \gamma_t^2}
\tag{25}
$$

$$
\leq e^{-T\gamma^2/8}
\tag{26}
$$

$$
\leq \frac{1}{m \cdot k},
\tag{27}
$$

where equation (25) follows by applying the approximation $1 + r \le e^r$ for all real $r$, equation (26) follows by Lemma 2, where $\epsilon = \gamma/(2k)$, and thus $\gamma/2 \le \gamma_t$ for all $t$. Equation (27) follows by plugging $T$. Since $\tilde{\mathcal{D}}_1$ is uniform over all examples and incorrect labels, then if the label-weighted training error of the combined classifier $\bar{h}_T$, which is always an integer multiple of $1/m(k-1)$, is at most $1/(m \cdot k)$, then the training error must in fact be zero. $\qquad\square$

# B   Omitted proofs of Section 4.2

**Lemma 9.** *Let $\mathcal{W}$ be an agnostic-PAC learner for the base hypothesis class $\mathcal{H} \subseteq \mathcal{Y}^{\mathcal{X}}$. Let $\gamma > 0$, and training set $S$ of $m$ labelled examples. If $\mathcal{H}$ is empirically $\gamma$-weak learnable with respect to $S$, then applying Algorithm 1 with $T = \frac{8(\log(m)+\log(k))}{\gamma^2}$, outputs $\bar{h}_T$ that is consistent with $S$. Furthermore, the empirical margin of $\bar{h}_T$ is at least $\gamma/8$.*

*Proof.* In this proof we use the notation introduced in the proof of Theorem 3. The consistency statement of the Lemma follows by Theorem 3. Next, we show that the empirical probability of $\bar{h}_T$ to have a low margin is zero. Specifically we show that,

$$F(x,y) - \max_{\ell \neq y} F(x,\ell) \ge \bar{\alpha} \cdot \gamma/8,$$

holds for all $(x,y) \in S$, where $\bar{\alpha} = \sum_{t=1}^{T} \alpha_t$ denotes the normalizing factor of $F$. Observe that the converse event occurs if and only if, for some $(x,y) \in S$,

$$\exp\left( -F(x,y) + \max_{\ell \neq y} F(x,\ell) + \bar{\alpha} \cdot \gamma/8 \right) \ge 1,$$

which in turn occurs if,

$$1 \le e^{\bar{\alpha}\gamma/8} \max_{\ell \neq y} e^{F(x,\ell)-F(x,y)} \le e^{\bar{\alpha}\gamma/8} \sum_{\ell \neq y} e^{F(x,\ell)-F(x,y)} = e^{\bar{\alpha}\gamma/8} \sum_{\ell \neq y} e^{F(x,\ell)-F(x,y)}. \qquad (28)$$

Then, we get that,

$$\mathbb{P}_{(x,y)\sim S}\left[ F(x,y) \text{ has margin} \le \gamma/8 \right] = \frac{1}{m} \sum_{i=1}^{m} \mathbf{1}\left[ F(x,y) - \max_{\ell \neq y} F(x,\ell) \le \bar{\alpha} \cdot \gamma/8 \right] \qquad (29)$$

$$\le \frac{e^{\bar{\alpha}\gamma/8}}{m} \sum_{i=1}^{m} \sum_{\ell \neq y} e^{F(x,\ell)-F(x,y)} \qquad (30)$$

$$= e^{\bar{\alpha}\gamma/8}(k-1)\frac{1}{m(k-1)} \sum_{i=1}^{m} \sum_{\ell \neq y} e^{F(x,\ell)-F(x,y)} \qquad (31)$$

$$= e^{\bar{\alpha}\gamma/8}(k-1)\prod_{t=1}^{T} Z_t, \qquad (32)$$

$$\qquad (33)$$

where the first inequality follows by (28), and the last equality uses Equation (18) (and summing $\tilde{\mathcal{D}}_{T+1}$ to 1 as in (19)). Recall that, as in the proof of Theorem 3, applying Lemma 2 with a choice of $\epsilon^w = \gamma/(2k)$, we get that $\gamma/2 \le \gamma_t$ for all $t$. Thus, by that, and by the definition of $\bar{\alpha}$, and the bound on $Z_t$ (see (23)), we get,

$$\mathbb{P}_{(x,y)\sim S}\left[ F(x,y) \text{ has margin} \le \gamma/8 \right] \le (k-1)\prod_{t=1}^{T} e^{\alpha_t \gamma_t/4} Z_t, \qquad (34)$$

$$= (k-1)\prod_{t=1}^{T}(1-\gamma_t)^{\frac{1}{2}-\frac{\gamma_t}{4}}(1+\gamma_t)^{\frac{1}{2}+\frac{\gamma_t}{4}} \qquad (35)$$

$$\le (k-1)\left[(1-\gamma/2)^{1-\frac{\gamma}{4}}(1+\gamma/2)^{1+\frac{\gamma}{4}}\right]^{T/2} \qquad (36)$$

$$\leq (k-1)\, e^{-T\gamma^2/8}, \tag{37}$$

where the first inequality holds since the inner expression of the product is a decreasing function of $\gamma_t$, and (36) holds since $\gamma/2 \leq \gamma_t$. Equation (37) follows by the the fact that the following inequality holds for all $\gamma \in (0,1)$,

$$-\frac{\gamma^2}{2\big((1-\frac{\gamma}{4})\ln(1-\frac{\gamma}{2}) + (1+\frac{\gamma}{4})\ln(1+\frac{\gamma}{2})\big)} < 8$$

Lastly, by setting $T$ as in the Lemma and plugging into (37), we get,

$$\mathbb{P}_{(x,y)\sim S}\Big[F(x,y) \text{ has margin} \leq \gamma/8\Big] \leq \frac{k-1}{m \cdot k}, \tag{38}$$

and since the empirical probability must be a multiple of $\frac{1}{m}$, we get that it must in fact be zero. $\quad\square$

## B.1 Definitions needed for the following proofs

For the sake of the next proofs, we define the following notions of combined hypotheses. Define $\overline{\mathcal{H}} \subseteq [0,1]^{\mathcal{X}\times\mathcal{Y}}$ be the set of all weighted hypotheses from $\mathcal{H}$, i.e.,

$$\overline{\mathcal{H}} = \Big\{ \bar{h} : (x,\ell) \mapsto \sum_{h\in\mathcal{H}} \lambda(h) \cdot \mathbb{1}[h(x)=\ell] \,\Big|\, \lambda \in \Delta_{\mathcal{H}} \Big\}. \tag{39}$$

Observe that taking $\arg\max_\ell \bar{h}(x,\ell)$ corresponds to the plurality-vote prediction (see Equation (1)). Furthermore, observe that there exists a combined hypotheses $\bar{h} \in \overline{\mathcal{H}}$ which corresponds to the weighted average $\sum_{t=1}^{T} \alpha_t \cdot \mathbb{1}[h_t(x)=\ell]$, obtained by the booster, in Algorithm 1 (and by taking its plurality-vote prediction we get the final output predictor $\bar{h}_T(x)$). Assume that $\alpha = (\alpha_1, ..., \alpha_T)$ is the normalized weight vector of the weights obtained by the algorithm. We overload notation by denoting this combined, weighted, hypothesis by $\bar{h}_T(x,\ell)$. Next, we define $\overline{\mathcal{H}}_n \subseteq [0,1]^{\mathcal{X}\times\mathcal{Y}}$ as the set of all underlined{unweighted} combined hypotheses of at most $n$ elements,

$$\overline{\mathcal{H}}_n = \Big\{ \bar{h} : (x,\ell) \mapsto \frac{1}{n}\sum_{j=1}^{n} \mathbb{1}[h_j(x)=\ell] \,\Big|\, h_1, ..., h_n \in \mathcal{H} \Big\}. \tag{40}$$

Note that the same $h \in \mathcal{H}$ may appear multiple times in such a combined hypothesis. The main idea of the proof is to approximate the (weighted) combined hypotheses $\bar{h}_T$, by randomly polling its constituents from its corresponding distribution $\alpha$. Towards that end, we prove a useful variation of the uniform convergence property applied to the margin of the combined hypotheses. Denote the margin of a combined hypotheses $f$, for a fixed $(x,y) \in \mathcal{X}\times\mathcal{Y}$, as follows,

$$\sigma(f;x,y) = f(x,y) - \max_{\ell\neq y} f(x,\ell). \tag{41}$$

The $\sigma$ uniform convergence bound given in Lemma 11 is based on the following combinatorial result.

**Lemma 10.** *[14] For every hypothesis class $F \subseteq \mathcal{Y}^{\mathcal{X}}$, $|F| \leq |\mathcal{X}|^{d_N(F)}|\mathcal{Y}|^{2d_N(F)}$.*

**Lemma 11** ($\sigma$ uniform convergence). *Let $\mathbf{D}$ be a distribution over $\mathcal{X}\times\mathcal{Y}$ that is $\gamma$-realizable for $\mathcal{H}$ and $\gamma > 0$, and let $S$ be a training set of $m$ i.i.d. samples from $\mathbf{D}$. Let $\delta > 0$. Then, with probability $1-\delta$ over the random choice of $S$, for all $n \geq 1$, $\bar{h} \in \overline{\mathcal{H}}_n$,*

$$\mathbb{P}_{\mathbf{D}}\Big[\sigma(\bar{h};x,y) \leq \frac{\gamma}{2}\Big] \leq \mathbb{P}_S\Big[\sigma(\bar{h};x,y) \leq \frac{\gamma}{2}\Big] + \epsilon_n,$$

*where $\epsilon_n = \sqrt{32\big(4dn\ln(mk) + \ln(8n(n+1)/\delta)\big)/m}$, and $d = d_N(\mathcal{H})$.*

*Proof.* Let $\mathcal{Z} = \mathcal{X}\times\mathcal{Y}$, and for any $\bar{h} \in \overline{\mathcal{H}}_n$, define a *low-margin* subset by,

$$\mathcal{Z}_{\bar{h}} = \Big\{ (x,y) \in \mathcal{Z} : \sigma(\bar{h};x,y) \leq \gamma/2 \Big\}.$$

Observe that any $\bar{h} \in \overline{\mathcal{H}}_n$ is determined by some $h_1, \ldots, h_n \in \mathcal{H}$, and that the restriction $\bar{h}|_Z$ of $\bar{h}$ to subset $Z$ is determined by the restrictions $h_1|_Z, \ldots, h_n|_Z$. Denote the collection of all low-margin subsets of $\mathcal{Z}$ by, $F_n = \left\{ \mathcal{Z}_{\bar{h}} : \bar{h} \in \overline{\mathcal{H}}_n \right\}$. Define the *growth function* of $F_n$ as,

$$\Pi_{F_n}(m) = \sup \left\{ |\{Z \cap Z' : Z' \in F_n\}| : Z \in \mathcal{Z}^m \right\}.$$

That is, $\Pi_{F_n}(m)$ is the maximal number of "in-out" behaviors (dichotomies) realizable by low-margin subsets (sets in $F_n$) on a finite set of $m$ points. Observe that the number of such dichotomies is determined by the $\sigma$ margin behaviors of all $\bar{h} \in \overline{\mathcal{H}}_n$, which in turn can be captured by 2 values for each $\bar{h}$ per example, (i.e., $\bar{h}(x, y)$ and $\max_{\ell \neq y} \bar{h}(x, \ell)$). Each $\bar{h}$ is in turn determined by some choice of $h_1, \ldots, h_n \in \mathcal{H}$. Using these observations we bound the growth function,

$$\Pi_{F_n}(m) = \sup_{Z \in \mathcal{Z}^m} \left| \left\{ \langle \mathbf{1}[z_1 \in Z'], \ldots, \mathbf{1}[z_m \in Z'] \rangle : Z' \in F_n \right\} \right|$$

$$= \sup_{Z \in \mathcal{Z}^m} \left| \left\{ \langle \ldots, \mathbf{1}[\sigma(\bar{h}; x_i, y_i) \leq \gamma/2], \ldots \rangle : \bar{h} \in \overline{\mathcal{H}}_n \right\} \right|$$

$$\leq \sup_{Z \in \mathcal{Z}^m} \left| \left\{ \langle \bar{h}(x_1, y_1), \ldots, \bar{h}(x_m, y_m) \rangle : \bar{h} \in \overline{\mathcal{H}}_n \right\} \right|^2$$

$$\leq \sup_{Z \in \mathcal{Z}^m} \left| \left\{ \langle \mathbf{1}[h(x_1) = y_1], \ldots, \mathbf{1}[h(x_m) = y_m] \rangle : h \in \mathcal{H} \right\} \right|^{2n}$$

$$= \sup_{Z \in \mathcal{Z}^m} |\mathcal{H}|_Z|^{2n} \leq m^{2dn} k^{4dn},$$

where the last inequality follows by Lemma 10. Next, we employ a bound given by a generalized form of uniform-convergence, as in Theorem 2.6, [16], which rather than showing that the training error generalizes to the population loss, shows the characterization of how any property, captured by subsets of the space $\mathcal{Z}$, generalizes. In particular, we get that for $n \geq 1$,

$$\mathbb{P}\left[\exists \mathcal{Z}_{\bar{h}} \in F_n, \; \mathbb{P}_{z \sim \mathcal{D}}[z \in \mathcal{Z}_{\bar{h}}] \geq \mathbb{P}_{z \sim S}[z \in \mathcal{Z}_{\bar{h}}] + \epsilon_n \right] \leq 8\Pi_{F_n}(m)e^{-m\epsilon_n^2/32}.$$

Therefore, by plugging in $\epsilon_n$, we get that the statement in the Lemma occurs with probability at least $1 - \delta/(n(n+1))$, for all $\bar{h} \in \overline{\mathcal{H}}_n$. By the union bound, this same statement holds for all $n \geq 1$ simultaneously with probability at least $1 - \delta$, proving the lemma. $\qquad \square$

## B.2 Proof of Theorem 6

*Proof.* In this proof we consider the notation introduced above in Section B.1. First, consider a *fixed* training set $S$, and let $\bar{h}_T$ be the predictor outputted by Algorithm 1 applied to $S$. We approximate $\bar{h}_T$ by sampling $\tilde{h}_j \sim \alpha$ (i.e., set $\tilde{h}_j := h_t$ w.p. $\alpha_t$), i.i.d. for each $j = 1 \ldots n$, and set,

$$\tilde{h}(x, \ell) = \frac{1}{n} \sum_{j=1}^{n} \mathbb{1}[\tilde{h}_j(x) = \ell].$$

Observe that $\tilde{h} \in \overline{\mathcal{H}}_n$ (defined in Equation (40)). In this proof, we will use $\tilde{h}$ to approximate $\bar{h}_T$. Towards that end, we first show that for a fixed $x \in \mathcal{X}$, $\ell \in [k]$, $\gamma' > 0$, $n \geq 1$, we have:

$$\mathbb{P}_{\tilde{h}}\left[ \left| \tilde{h}(x, \ell) - \bar{h}_T(x, \ell) \right| \geq \gamma'/2 \right] \leq 2e^{-n\gamma'^2/2}. \tag{42}$$

This holds since by the definition of $\tilde{h}$ we have $\mathbb{E}_{\tilde{h}}[\tilde{h}(x, \ell)] = \bar{h}_T(x, \ell)$, and by applying Hoeffding's inequality we obtain Equation (42). Next, we show further that the *margins* of $\bar{h}_T$ and of $\tilde{h}$ are close. Specifically, we consider the $\sigma$ notation in Equation (41), and show that for any fixed $x, y$, and for any $\gamma' > 0$, $n \geq 1$, we have,

$$\mathbb{P}_{\tilde{h}}\left[ \left| \sigma(\tilde{h}; x, y) - \sigma(\bar{h}_T; x, y) \right| \geq \gamma'/2 \right] \tag{43}$$

$$= \mathbb{P}_{\tilde{h}}\left[ \left| \left( \tilde{h}(x, y) - \max_{\ell \neq y} \tilde{h}(x, \ell) \right) - \left( \bar{h}_T(x, y) - \max_{\ell \neq y} \bar{h}_T(x, \ell) \right) \right| \geq \gamma'/2 \right] \tag{44}$$

$$\leq \mathbb{P}_{\tilde{h}}\left[\left|\tilde{h}(x,y) - \bar{h}_T(x,y)\right| + \max_{\ell \neq y}\left|\tilde{h}(x,\ell) - \bar{h}_T(x,\ell)\right| \geq \gamma'/2\right] \tag{45}$$

$$\leq 2\,\mathbb{P}_{\tilde{h}}\left[\max_{\ell \neq y}\left|\tilde{h}(x,\ell) - \bar{h}_T(x,\ell)\right| \geq \gamma'/4\right] \tag{46}$$

$$\leq 2\sum_{\ell \neq y}\mathbb{P}_{\tilde{h}}\left[\left|\tilde{h}(x,\ell) - \bar{h}_T(x,\ell)\right| \geq \gamma'/4\right]. \tag{47}$$

Then, by combining the above with (42), we get that for any $D$ distribution over $\mathcal{X} \times \mathcal{Y}$, and for any $\gamma' > 0$, $n \geq 1$, we have,

$$\mathbb{E}_D\left[\mathbb{P}_{\tilde{h}}\left[|\sigma(\tilde{h};x,y) - \sigma(\bar{h}_T;x,y)| \geq \gamma'/2\right]\right] \leq \mathbb{E}_D\left[4(k-1)e^{-n\gamma'^2/2}\right] = 4(k-1)e^{-n\gamma'^2/2}. \tag{48}$$

Next, we apply the bound in (48) to distribution $\mathbf{D}$, and get that,

$$\mathbb{P}_{\mathbf{D}}\left[\sigma(\bar{h}_T;x,y) \leq 0\right] = \mathbb{P}_{\mathbf{D},\tilde{h}}\left[\sigma(\bar{h}_T;x,y) \leq 0\right] \tag{49}$$

$$\leq \mathbb{P}_{\mathbf{D},\tilde{h}}\left[\sigma(\tilde{h};x,y) \leq \frac{\gamma'}{2}\right] + \mathbb{P}_{\mathbf{D},\tilde{h}}\left[\sigma(\bar{h}_T;x,y) \leq 0 \bigwedge \sigma(\tilde{h};x,y) > \frac{\gamma'}{2}\right] \tag{50}$$

$$\leq \mathbb{P}_{\mathbf{D},\tilde{h}}\left[\sigma(\tilde{h};x,y) \leq \frac{\gamma'}{2}\right] + \mathbb{P}_{\mathbf{D},\tilde{h}}\left[\left|\sigma(\tilde{h};x,y) - \sigma(\bar{h}_T;x,y)\right| > \frac{\gamma'}{2}\right] \tag{51}$$

$$\leq \mathbb{P}_{\mathbf{D},\tilde{h}}\left[\sigma(\tilde{h};x,y) \leq \frac{\gamma'}{2}\right] + 4(k-1)e^{-n\gamma'^2/2}, \tag{52}$$

where the first inequality follows from the the simple fact that for any two events $a$ and $b$, $\mathbb{P}[a] = \mathbb{P}[a \wedge b] + \mathbb{P}[a \wedge \neg b] \leq \mathbb{P}[b] + \mathbb{P}[a \wedge \neg b]$. Using a similar derivation, applied to the empirical distribution that is uniform over $S$, we obtain,

$$\mathbb{P}_{S,\tilde{h}}\left[\sigma(\tilde{h};x,y) \leq \frac{\gamma'}{2}\right] \leq \mathbb{P}_{S,\tilde{h}}\left[\sigma(\bar{h}_T;x,y) \leq \gamma'\right] + \mathbb{P}_{S,\tilde{h}}\left[\sigma(\tilde{h};x,y) \leq \frac{\gamma'}{2} \bigwedge \sigma(\bar{h}_T;x,y) > \gamma'\right] \tag{53}$$

$$\leq \mathbb{P}_{S,\tilde{h}}\left[\sigma(\bar{h}_T;x,y) \leq \gamma'\right] + \mathbb{P}_{S,\tilde{h}}\left[\left|\sigma(\tilde{h};x,y) - \sigma(\bar{h}_T;x,y)\right| > \frac{\gamma'}{2}\right] \tag{54}$$

$$\leq \mathbb{P}_{S,\tilde{h}}\left[\sigma(\bar{h}_T;x,y) \leq \gamma'\right] + 4(k-1)e^{-n\gamma'^2/2}. \tag{55}$$

Lastly, by applying Lemma 11 and using the $\sigma$ uniform-convergence property, we get that with probability at least $1 - \delta$ over the random choice of $S$, for any $n \geq 1$, and $\bar{h}_n \in \overline{\mathcal{H}}_n$,

$$\mathbb{P}_{\mathbf{D}}\left[\sigma(\bar{h}_n;x,y) \leq \frac{\gamma'}{2}\right] \leq \mathbb{P}_S\left[\sigma(\bar{h}_n;x,y) \leq \frac{\gamma'}{2}\right] + \epsilon_n, \tag{56}$$

where $\epsilon_n = O\left(\sqrt{\frac{dn\log(mk) + \log(1/\delta)}{m}}\right)$, with $d = d_N(\mathcal{H})$ denoting the Natarajan dimension of $\mathcal{H}$.

Observe that by using marginalization, we get that the above bound holds with respect to a random $\tilde{h}$, rather than a fixed $\bar{h}_n$. Then, combining the above Equations (52), then (56) (using marginalization), then (55), we get that with probability $1 - \delta$,

$$\mathbb{P}_{\mathbf{D}}\left[\sigma(\bar{h}_T;x,y) \leq 0\right] \leq \mathbb{P}_{\mathbf{D},\tilde{h}}\left[\sigma(\tilde{h};x,y) \leq \frac{\gamma'}{2}\right] + 4e^{-n\gamma'^2/2} \tag{57}$$

$$\leq \mathbb{P}_{S,\tilde{h}}\left[\sigma(\tilde{h};x,y) \leq \frac{\gamma'}{2}\right] + 4e^{-n\gamma'^2/2} + \epsilon_n \tag{58}$$

$$\le \mathbb{P}_{S,\tilde{h}}\left[\sigma(\bar{h}_T; x, y) \le \gamma'\right] + 8e^{-n\gamma'^2/2} + \epsilon_n. \tag{59}$$

By setting $n = \frac{2}{\gamma^2}\ln(\frac{16}{\epsilon})$, we get that $8e^{-n\gamma'^2/2} = \epsilon/2$. Then, by setting $m$ as stated in the Theorem, we get that with probability $1 - \delta$, the overall bound is,

$$\mathbb{P}_{\mathbf{D}}\left[\sigma(\bar{h}_T; x, y) \le 0\right] \le \mathbb{P}_S\left[\sigma(\bar{h}_T; x, y) \le \gamma'\right] + \epsilon. \tag{60}$$

Lastly, by Lemma 9, we get that the empirical margin probability is zero, for $\gamma' = \gamma/8$, which completes the proof. $\qquad\square$

## C  Lower Bound: Omitted proofs of Section 5

The proof of Theorem 7 will occupy this section. Although the intuitive idea outlined in Section 5 is simple, there are many subtle but technical details that will need to be worked out to ensure that all of the formal requirements of the learning model are satisfied. We consider the model definition as detailed in Section 5, and describe the construction of the base class $\mathcal{H}$ below.

### C.1  The construction of $\mathcal{H}$

Let $\mathcal{X}$ be the finite domain, with $n = \frac{1}{2\gamma}$ data points, and assume that $4n < k$. We fix the constant function $f(x) = 1$ to be the target labelling. Then, the labelled data is $\{(x_1, 1)...(x_n, 1)\}$, and let the target distribution $\mathcal{D}$ be the uniform distribution over it. We note that our construction and the lower bound argument hold even for the case that the boosting algorithm has full access to the data and distribution $\mathcal{D}$. However, recall that it is restricted to only output a plurality-vote of hypotheses $h^1, ...h^T \in \mathcal{H}$ obtained via $\mathcal{W}$. Towards defining the base class $\mathcal{H}$, consider hypotheses of the following form; for all $j \in [n]$ and $\ell \in \{2, ..., k\}$, define

$$h_{j,\ell}(x) = \begin{cases} 1 & \text{if } x = x_j, \\ \ell & \text{otherwise.} \end{cases} \tag{61}$$

We denote the set of all such hypotheses as $\mathcal{H}_{all}$. Next, we describe the desired properties of the constructed base class. For any hypothesis class $\mathcal{H} \subset \mathcal{H}_{all}$ of size $|\mathcal{H}| = 2n-1$, consider the following two conditions:

(I) For each index $j = 1, ..., n-1$ there are exactly 2 distinct labels $\ell_1, \ell_2 \in [k]$ such that $h_{j,\ell_1}, h_{j,\ell_2} \in \mathcal{H}$. For index $j = n$, there exists exactly 1 label $\ell$ such that, $h_{n,\ell} \in \mathcal{H}$.

(II) For each label $\ell = 2, ..., k$, there is at most 1 index $j \in [n]$ such that $h_{j,\ell}$ in $\mathcal{H}$.

**Lemma 12.** *Let $\mathcal{H} \subset H_{all}$ be a hypothesis class of size $2n-1$, such that both (I), (II) hold. Then, the distribution $\mathcal{D}$, uniform over $\mathcal{X} \times \{1\}$, is $\gamma/2$-realizable w.r.t. $\mathcal{H}$ (see Definition 1).*

*Moreover, if only condition (I) fails such that there are $q$ indices $j \in [n]$ for each of which there is at most 1 hypothesis of the form $h_{j,\ell}$ in $\mathcal{H}$, then any plurality-vote over $\mathcal{H}$ incurs error of at least $2\gamma(q-1) = \frac{q-1}{n}$.*

We use the probabilistic method to construct the base hypothesis class $\mathcal{H}$, a subset of $H_{all}$ of size $2n - 1$. Specifically, we let the class $\mathcal{H}$ be entirely fixed, apart from a single random hypothesis, $h_{n-1,L}$, where $L$ denotes a random variable drawn uniformly at random from the set of labels $2n, ..., k$. In particular, the class $\mathcal{H} \subset \mathcal{H}_{all}$ is constructed as follows. For each index $j < n - 1$, we have exactly 2 fixed hypotheses $h_{j,2j}$ and $h_{j,2j+1}$ in $\mathcal{H}$; for $j = n - 1$, we have one fixed hypotheses $h_{n-1,2n-2}$, and the other is the random hypotheses $h_{n-1,L}$. Lastly, for $j = n$ we have a single fixed

hypotheses $h_{n,2n-1}$. The overall class is:

$$
\begin{aligned}
\mathcal{H} = \big\{ & h_{1,2}, h_{1,3}, \\
& h_{2,4}, h_{2,5}, \\
& h_{3,6}, h_{3,7}, \\
& \quad\vdots \\
& h_{n-3,2n-6}, h_{n-3,2n-5} \\
& h_{n-2,2n-4}, h_{n-2,2n-3} \\
& h_{n-1,2n-2}, h_{n-1,L} \\
& h_{n,2n-1} \big\}.
\end{aligned}
\tag{62}
$$

Observe that the class $\mathcal{H}_{all}$ is an easy-to-learn class, and it be easily verified that it has a constant Natarajan dimension of 2. Since $\mathcal{H} \subset \mathcal{H}_{all}$ for any $L$, we get that $\mathcal{H}$ has a constant Natarajan dimension as well. By Lemma 12, there exists a plurality-vote over $\mathcal{H}$ that is consistent with the data, with margin $\gamma/2$. That is, the distribution $\mathcal{D}$ which is uniform over $\mathcal{X} \times \{1\}$ is $\gamma/2$-realizable, with respect to $\mathcal{H}$.

We refer to the randomly chosen hypothesis $h_{n-1,L}$ as the *hidden* hypothesis. Note that if the booster cannot "discover" the hidden hypothesis throughout its run (i.e., obtain it via the weak learner), then by Lemma 12, it will fail to achieve a low generalization error (with error of at least $2\gamma$). Next, we show that indeed it will fail to obtain the hidden hypothesis.

### C.2 The "finder-chooser" game

Intuitively, applying any boosting algorithm on the random class $\mathcal{H}$ constructed above requires the booster to find the hidden hypothesis, which is a challenging task. This is the crucial observation needed for proving the bound, and is demonstrated in the simpler setting of a "finder-chooser" game, described next. Consider a setting of an iterative game in which a *"chooser"* player picks a "hidden" element $L$ out of a set of elements $K$, of size $|K| = k$, and a *"finder"* player iteratively attempts to discover it. Specifically, the finder is only required to find any small subset of elements from $K$ that the hidden element $L$ belongs to. Towards that end, in each round the finder queries the chooser, and the chooser replies with a binary response, which indicates if the chooser has indeed picked the hidden element. Concretely, we consider 2 different query-models in which the finder is allowed to interact with the chooser; via weights, or via examples. In either case, the finder is given a binary response in each round, of whether or not the hidden element is in the subset it picked. If the finder failed, it continues to the next round, otherwise it succeeded to find a subset which contains $L$, and the game ends. It is clear that there is a trade-off between the size of the picked subsets and the number of rounds needed to find the containing subset. The Lemmas below formalize this trade-off in both models, and give bounds on the probability that the finder succeeds, when the chooser is uniform over $K$.

#### C.2.1 The "finder-chooser" game: Weighting model

Let $k, T > 0$, and fix a threshold $\theta \in (0,1)$. The model is defined as follows. First, the element $L$ is sampled uniformly over the set $[k] = \{1, ..., k\}$. Next, for each round $t = 1...T$, let $D_t \in \Delta_k$ be some distribution of $[k]$ determined iteratively as the computation output of a "finder" algorithm $F$, defined as follows. Formally, the finder $F$ is a fixed and deterministic function of its previous picks $D_{t'}$, and previous binary responses $I_{t'}$, for all $t' < t$, and randomness source $\mathcal{R}$, where $I_t = \mathbf{1}\left[ D_t(L) \geq \theta \right]$.

**Lemma 13** (**Weights**). *For any finder $F$ algorithm in the the weighting model, it holds that,*

$$
\mathbb{P}_{L,\mathcal{R}}\left[ \exists\, t \leq T, \ D_t(\ell) \geq \theta \right] \leq \frac{T}{\theta k}.
$$

#### C.2.2 The "finder-chooser" game: Example model

We now consider a slight modification of the setting defined above. Let $T, m, k > 0$. The model is defined as follows. As in the previous model, the element $L$ is sampled uniformly over the set

$[k] = \{1, ..., k\}$. Next, for each round $t = 1...T$, let $\mathcal{L}_t \subset K$, be a subset of at most $m$ elements, that is determined iteratively via a "finder" algorithm $F$, defined as follows. Formally, the finder $F$ is a fixed and deterministic function of its previous picks $\mathcal{L}_{t'}$, and previous binary responses $I_{t'}$, for all $t' < t$, and randomness source $\mathcal{R}$, where $I_t = \mathbf{1}\left[L \in \mathcal{L}_t\right]$.

**Lemma 14** (**Examples**). *For any finder $F$ algorithm in the the example model, it holds that,*

$$\mathbb{P}_{L, \mathcal{R}}\left[\exists\, t \leq T, \;\; L \in \mathcal{L}_t\right] \leq \frac{Tm}{k}.$$

Next, we are ready to prove the main result, stated in Theorem 7.

## C.3   Proof of Theorem 7

*Proof.* Consider the randomly-constructed class $\mathcal{H}$ in Equation (62), and the distribution $\mathcal{D}$ that is uniform over $\mathcal{X} \times \{1\}$, which as shown above is $\gamma/2$-realizable for $\mathcal{H}$ (see Definition 1). We use the probabilistic method to complete the proof, for any boosting algorithm $B$ (see the model definition in subsection 5.1).

There are several sources of randomness that are part of either the learning process or of our construction, namely the class $\mathcal{H}$ (in particular, the label $L$), the initial training set $S$, the internal randomness of the booster $\mathcal{R}$.

We will show that with respect to all of the sources of randomness, $B$'s error is probable to be large. That is, we show that,

$$\mathbb{P}_{\mathcal{H}, S, \mathcal{R}}\left[\mathbb{P}_{\mathcal{D}}\left[\bar{h}_B(x) \neq y\right] \leq \epsilon^B\right] < 1 - \delta^B. \tag{63}$$

This is sufficient for the proof since it implies, $\mathbb{P}_{\mathcal{H}, S, \mathcal{R}}\left[\mathbb{P}_{\mathcal{D}}\left[\bar{h}_B(x) \neq y\right] > \epsilon^B\right] \geq \delta^B$, which is equivalent by marginalization to, $\mathbb{E}_{\mathcal{H}}\left[\mathbb{P}_{S, \mathcal{R}}\left[\mathbb{P}_{\mathcal{D}}\left[\bar{h}_B(x) \neq y\right] > \epsilon^B \big| \mathcal{H}\right]\right] \geq \delta^B$. This in turn implies that there exists a *particular* class $\mathcal{H}$ for which, $\mathbb{P}_{S, \mathcal{R}}\left[\mathbb{P}_{\mathcal{D}}\left[\bar{h}_B(x) \neq y\right] > \epsilon^B \big| \mathcal{H}\right] \geq \delta^B$, as claimed. Therefore, it is sufficient to prove Equation (63). Next, we give a weak learner $\mathcal{W}$ for both models of Boosting-by-reweighting and Boosting-by-resampling, for which (63) holds.

**Boosting-by-reweighting**   Let the learner $\mathcal{W}$ be defined as follows: for any $t$, if $\epsilon_t^w > \theta$, and there exists any $\epsilon_t^w$-optimal hypothesis with respect to $\mathcal{D}_t$ that is not the hidden hypothesis, then $\mathcal{W}$ returns it to $B$. Otherwise, $\mathcal{W}$ returns the hidden hypothesis. Denote $\mathcal{H}_T$ as the set of all hypotheses observed by the booster up to round $T$. Denote $OPT_t \subset \mathcal{H}$ as the set of all $\epsilon_t^w$-optimal hypotheses with respect to $\mathcal{D}_t$. Then, we upper bound the probability that $B$ succeeds to have low error,

$$
\begin{aligned}
\mathbb{P}_{\mathcal{H}, S, \mathcal{R}}\left[\mathbb{P}_{\mathcal{D}}\left[\bar{h}_B(x) \neq y\right] \leq \epsilon^B\right] &\leq \mathbb{P}_{L, S, \mathcal{R}}\left[h_{n-1, L} \in \mathcal{H}_T\right] && \text{(by Lemma 12)} \\
&\leq \mathbb{P}_{L, S, \mathcal{R}}\left[\exists\, t \leq T, \; OPT_t = \{h_{n-1, L}\}\right] && \text{(by definition of } \mathcal{W}) \\
&\leq \mathbb{P}_{L, S, \mathcal{R}}\left[\exists\, t \leq T, \; D_t(L) > \epsilon_t^w\right] && \text{(by Lemma 15)} \\
&\leq \mathbb{P}_{L, S, \mathcal{R}}\left[\exists\, t \leq T, \; D_t(L) > \theta\right], && \text{(by assumption on } \epsilon_t^w)
\end{aligned}
$$

where $D_t$ denotes the marginal distribution over labels of $\mathcal{D}_t$, i.e., $D_t(\ell) = \mathbb{P}_{(x, y) \sim \mathcal{D}_t}[y = \ell]$.

We next describe a reduction to the "finder-chooser" setting defined in Lemma 13. In more detail, we will show how to convert the given boosting algorithm $B$, to a finder's strategy as defined in C.2.1. Assume we are given a boosting algorithm $B$ as above, and consider an instance of the finder-chooser game which we are going to solve using access to $B$. Towards this end, define the base class $\mathcal{H}$ as in Equation (62), where the label $L$ is determined by the secret label held by the chooser. Observe that the probability in the last inequality above, is equivalent to the probability that the finder succeeds to "win" the game. The reduction is obtained by substituting the sequence weights generated by the

finder, with the marginal distributions $D_1, ..., D_T$ generated by the booster, where the set $K$ and size $k$ in Lemma 13 corresponds to the set of labels $2n...k$, and size $k-2n-1$.

In particular, whenever the boosting algorithm submits a distribution $\mathcal{D}_t$ to the weak learner, our finder submits the corresponding marginal distribution $D_t$ to the chooser. Observe that the binary responses of the chooser correspond to the binary alternative cases for the booster $I_t = [D_t(L) \geq \theta]$.

Note that for the task of finding the hidden label $L$ (i.e., the probability that the booster finds the hidden hypothesis $h_{n-1,L}$), can by upper bounded by the setting in which it is only answered with binary responses, since no additional information of $L$ is gained by the booster for different responses of non-hidden hypotheses [4]. Then, we get,

$$\mathbb{P}_{\mathcal{H},S,\mathcal{R}}\left[\mathbb{P}_{\mathcal{D}}\left[\bar{h}_B(x) \neq y\right] \leq \epsilon^B\right] \leq \mathbb{P}_{L,S,\mathcal{R}}\left[\exists\, t \leq T,\ D_t(L) > \theta\right] \qquad \text{(by last inequality)}$$

$$\leq T \cdot \frac{1}{\theta} \cdot \frac{1}{k - 2n - 1} \qquad \text{(by Lemma 13)}$$

$$< 1/2 \leq 1 - \delta^B,$$

where the last inequality follows by the definition of $\theta$, and by the assumption that $\delta^B \leq 1/2$.

**Boosting-by-resampling** Let the learner $\mathcal{W}$ be an ERM learner such that for any $t$, when given $m_t^w < 1/\theta$ labelled examples, all of which are not labelled with the hidden label $L$, return any optimal hypothesis that minimizes the number of sample errors, other than the hidden hypothesis. Otherwise, return any optimal hypothesis. Note that the sample complexity required for the ERM $\mathcal{W}$ to agnostically-PAC learn $\mathcal{H}$ is at most $m_t^w(\epsilon, \delta) \leq O\left(\frac{\log(k) + \log(1/\delta)}{\epsilon^2}\right)$ (see Equation 7, and recall that $\mathcal{H}$ has a constant dimension). However, below we show that when $m_t^w$ is too small ($m_t^w < 1/\theta \approx O(k/T)$), then the booster $B$ fails. Denote $\mathcal{H}_T$ as the set of all hypotheses observed by the booster up to round $T$. Let $\mathcal{L}_t$ denote the set of all labels for which a labelled example was is fed to $\mathcal{W}$ at time $t$. Then, we upper bound the probability that $B$ succeeds to have low error,

$$\mathbb{P}_{\mathcal{H},S,\mathcal{R}}\left[\mathbb{P}_{\mathcal{D}}\left[\bar{h}_B(x) \neq y\right] \leq \epsilon^B\right] \leq \mathbb{P}_{L,S,\mathcal{R}}\left[h_{n-1,L} \in \mathcal{H}_T\right] \qquad \text{(by Lemma 12)}$$

$$\leq \mathbb{P}_{L,S,\mathcal{R}}\left[\exists\, t \leq T,\ L \in \mathcal{L}_t\right]. \qquad \text{(by definition of } \mathcal{W}\text{)}$$

Recall that the "hidden label" $L \sim \mathcal{U}(2n, ..., k)$ is chosen uniformly at random. As in the previous model, we apply the reduction to the "finder-chooser" setting defined in Lemma 14, by replacing the sequence of subsets $\mathcal{L}_1, ..., \mathcal{L}_T$ picked by the booster, with the subset of elements generated by the finder, where the set $K$ and size $k$ in Lemma 14 corresponds to the set of labels $2n...k$, and size $k-2n-1$. Then, we get,

$$\mathbb{P}_{\mathcal{H},S,\mathcal{R}}\left[\mathbb{P}_{\mathcal{D}}\left[\bar{h}_B(x) \neq y\right] \leq \epsilon^B\right] \leq T \cdot m \cdot \frac{1}{k - 2n - 1} < \frac{T/\theta}{k - 2n - 1} < 1/2 \leq 1 - \delta^B,$$

where the last inequalities follow by the assumption that $m_t^w < 1/\theta$ for all $t$, by the definition of $\theta$, and by the assumption that $\delta^B \leq 1/2$.

$\square$

**Lemma 15.** *Let $\mathcal{H}$ be defined as in Equation* (62)*, for some fixed label $L \in \{2n, ..., k\}$. Let $\mathcal{W}$ be an agnostic PAC learner for $\mathcal{H}$. Let $\mathcal{D}$ be any distribution over $\mathcal{X} \times \mathcal{Y}$, and $\epsilon^w > 0$. Then, if the only $\epsilon$-optimal hypothesis in $\mathcal{H}$ with respect to $\mathcal{D}$ is the hypothesis $h_{n-1,L}$, i.e., if it holds that,*

$$\max_{\substack{j \leq n \\ \ell < 2n}} \mathbb{P}_{\mathcal{D}}[h_{j,\ell}(x) = y] < \max_{h^* \in \mathcal{H}} \mathbb{P}_{\mathcal{D}}[h^*(x) = y] - \epsilon^w,$$

*then, for the corresponding marginal distribution over labels $D$, it holds that,*

$$D(L) = \sum_{j=1}^{n} \mathcal{D}(x_j, L) > \epsilon^w.$$

---

[4]That is, for different $\ell', \ell'' < 2n$, and any $t_1, t_0$ such that $t_1 > t_0$, it holds that $\mathbb{P}[D_{t_1}(L) \geq \theta | h_{t_0} = h_{j,\ell'}] = \mathbb{P}[D_{t_1}(L) \geq \theta | h_{t_0} = h_{j,\ell''}]$.

*Proof.* Observe that by the assumption, $h_{n-1,L} = \arg\max_{h^* \in \mathcal{H}} \mathbb{P}_{\mathcal{D}}[h^*(x) = y]$. Then, we have,

$$\mathcal{D}(x_{n-1}, 1) \leq \max_{j \leq n} \mathcal{D}(x_j, 1)$$

$$\leq \max_{\substack{j \leq n \\ \ell < 2n}} \left( \mathcal{D}(x_j, 1) + \sum_{j' \neq j} \mathcal{D}(x_{j'}, \ell) \right)$$

$$= \mathbb{P}_{\mathcal{D}}[h_{j,\ell}(x) = y]$$

$$< \mathbb{P}_{\mathcal{D}}[h_{n-1,L}(x) = y] - \epsilon^w$$

$$= \mathcal{D}(x_{n-1}, 1) + \sum_{j' \neq n-1} \mathcal{D}(x_{j'}, L) - \epsilon^w,$$

where the last inequality follows from the assumption. By re-arranging terms we get that,

$$\epsilon^w < \sum_{j' \neq n-1} \mathcal{D}(x_{j'}, L) \leq \sum_{j=1}^{n} \mathcal{D}(x_j, L) = D(L),$$

as claimed.

$\square$

### C.4 Proof of Lemma 12

*Proof.* We define $\lambda$ a distribution over the $2n-1$ hypotheses of a class $\mathcal{H}$ satisfying the conditions of the lemma, and show that it induces a consistent plurality-vote with margin $\gamma/2$. Denote each hypotheses in $\mathcal{H}$ by an index $i$, and assume w.l.o.g that hypothesis $i = 2n-1$ corresponds to the single hypotheses $h_{n,\ell}$ as defined in condition (I). Set $\xi = \frac{1}{4n}$. For all $i < 2n-1$, set $\lambda(i) = \frac{1-\xi}{2n-1}$, and set $\lambda(2n-1) = \frac{1}{2n-1} + \frac{2n-2}{2n-1}\xi$. Then, a point $x_j$, for any $j \leq n-1$, corresponds to exactly 2 hypotheses with indices $i_1, i_2$ that predict its label correctly, and the corresponding weight of that prediction is $\lambda(i_1) + \lambda(i_2)$, and any incorrect label will have weight of at most $\lambda(2n-1)$. For a point $x_n$, the corresponding correct and incorrect weights are $\lambda(2n-1)$ and $\lambda(i)$ for some $i < 2n-1$. Overall, the plurality-vote induced by $\lambda$ has a margin of,

$$\min\{2\lambda(i) - \lambda(2n-1), \lambda(2n-1) - \lambda(i)\} = \min\left\{ \frac{1-2n\xi}{2n-1}, \xi \right\}$$

$$= \min\left\{ \frac{1}{2} \cdot \frac{1}{2n-1}, \xi \right\}$$

$$\geq \frac{1}{4n} = \gamma/2.$$

To prove the second statement in the Lemma, assume that for a class $\mathcal{H}$ of size $2n - q$, condition (II) holds, but condition (I) fails, i.e., there are $j_1, ..., j_q \in [n]$ distinct indices such that for each such $j$, there is exactly 1 hypotheses of the form $h_{j,\ell} \in \mathcal{H}$, for some unique $\ell$, denoted $\ell_j$. Denote each hypothesis in $\mathcal{H}$ by an index $i = 1, ..., (2n-q)$, and assume w.l.o.g that for all $i \leq q$, the $i$-th hypothesis $h_i$ corresponds to $h_{j_i, \ell_{j_i}}$. Let $\lambda$ be any distribution over $\mathcal{H}$.

Then, let $i^* \in [2n-q]$ be the index of a hypothesis in $\mathcal{H}$, such that for all other $i \leq q$, $i \neq i^*$, $\lambda(i^*) > \lambda(i)$ (if the inequality is not strict, plurality-vote breaks ties arbitrarily, pick $i^*$ maximal by it). Since for each $x_i$ with $i \leq q$, the correct label 1 will get $\lambda(i)$ of the votes, whereas some incorrect label corresponding to hypothesis $i^*$ will get $\lambda(i^*)$ of the votes. If $i^* > q$, then for all $i \leq q$, the plurality-vote induced by $\lambda$ will err on $x_i$. Otherwise, if $i^* \leq q$, then it will err on exactly $q - 1$ such points.

Overall, the plurality-vote will err for at least $q - 1$ such $i$'s. Each such error corresponds to a portion of $1/n = 2\gamma$ of the population loss. Therefore, the plurality-vote induced by $\lambda$ over $\mathcal{H}$ incurs error of at least $2\gamma(q - 1)$.

$\square$

### C.5 Proof of Lemma 13

*Proof.* Let $\mathcal{I}_t = \mathbf{1}\left[\exists t' \leq t, I_{t'} = 1\right]$. Define $\mathcal{I}_0 = 0$. Fix an algorithm $F$ and its randomness, and observe that when simulated with any fixed sequence of binary responses, $I_1^F, ..., I_{t-1}^F$, it outputs

a sequence $v_1, ..., v_t$ that is deterministic. This follows by definition, as the output sequence $v_t$ is a deterministic function of the random bits of $F$ and the binary responses it observes $I_1^F, ..., I_{t-1}^F$, and if those are fixed, then so is the output. next, for any $t$, simulate $F$ with a response sequence of all zeros, $I_1^F = 0, ..., I_{t-1}^F = 0$, which then outputs $v_t$, and let $\mathcal{L}_t^{\mathbf{0}} = \{\ell \in [k] \mid D_t(\ell) \geq \theta\}$. First, observe that $|\mathcal{L}_t^{\mathbf{0}}| \leq 1/\theta$, and $|\cup_{t=1}^T \mathcal{L}_t^{\mathbf{0}}| \leq T/\theta$. Moreover, notice that for a fixed $F$, the set $\cup_{t=1}^T \mathcal{L}_t^{\mathbf{0}}$ is deterministic. Thus,

$$\mathbb{P}_{L,\mathcal{R}}\left[\exists\, t \leq T,\;\; D_t(L) \geq \theta\right] = \mathbb{P}_{L,\mathcal{R}}\left[\bigvee_{t=1}^T \mathcal{I}_{t-1} = 0 \wedge I_t = 1\right] \tag{64}$$

$$= \mathbb{E}_{\mathcal{R}}\left[\mathbb{P}_L\left[\bigvee_{t=1}^T \mathcal{I}_{t-1} = 0 \wedge I_t = 1 \middle| \mathcal{R}\right]\right] \tag{65}$$

$$= \mathbb{E}_{\mathcal{R}}\left[\mathbb{P}_L\left[\bigvee_{t=1}^T L \in \mathcal{L}_t^{\mathbf{0}} \middle| \mathcal{R}\right]\right] \tag{66}$$

$$= \mathbb{E}_{\mathcal{R}}\left[\mathbb{P}_L\left[L \in \bigcup_{t=1}^T \mathcal{L}_t^{\mathbf{0}} \middle| \mathcal{R}\right]\right] \tag{67}$$

$$\leq \frac{T}{\theta k}. \tag{68}$$

$\square$

### C.6   Proof of Lemma 14

*Proof.* Let $\mathcal{I}_t = \mathbf{1}\left[\exists t' \leq t,\; I_{t'} = 1\right]$. Define $\mathcal{I}_0 = 0$. Fix $\mathcal{R}$, and for any $t$, simulate $F$ with a response sequence of all zeros, $I_1 = 0, ..., I_{t-1} = 0$, which then outputs $v_t$, and let $\mathcal{L}_t^{\mathbf{0}}$ denote the set $\mathcal{L}_t$ obtained by the simulation. note that $|\cup_{t=1}^T \mathcal{L}_t^{\mathbf{0}}| \leq Tm$. Thus,

$$\mathbb{P}_{L,\mathcal{R}}\left[\exists\, t \leq T,\;\; \ell \in \mathcal{L}_t\right] = \mathbb{P}_{L,\mathcal{R}}\left[\bigvee_{t=1}^T \mathcal{I}_{t-1} = 0 \wedge I_t = 1\right] \tag{69}$$

$$= \mathbb{E}_{\mathcal{R}}\left[\mathbb{P}_L\left[\bigvee_{t=1}^T \mathcal{I}_{t-1} = 0 \wedge I_t = 1 \middle| \mathcal{R}\right]\right] \tag{70}$$

$$= \mathbb{E}_{\mathcal{R}}\left[\mathbb{P}_L\left[\bigvee_{t=1}^T \ell \in \mathcal{L}_t^{\mathbf{0}} \middle| \mathcal{R}\right]\right] \tag{71}$$

$$= \mathbb{E}_{\mathcal{R}}\left[\mathbb{P}_L\left[\ell \in \bigcup_{t=1}^T \mathcal{L}_t^{\mathbf{0}} \middle| \mathcal{R}\right]\right] \tag{72}$$

$$\leq \frac{Tm}{k}. \tag{73}$$

$\square$