# OpenReview forum: "Multiclass Boosting and the Cost of Weak Learning"
_NeurIPS.cc/2021/Conference — NeurIPS 2021 Poster_

### Official Review · Reviewer_i7mp · 2021-07-13

**Rating:** 7
**Confidence:** 2

**Summary:**

The authors propose  a new algorithm for multiclass boosting (Multiclass Adaboost) with strong theoretical guarantees. In particular, they show an upper bound by $O(\ln k)$ on the number of oracle calls needed for consistency on the training set, under a margin-realizability assumption, where $k$ is the number of possible outputs. Moreover, they provide an upper bound on the sample complexity of the algorithm in terms of the Natarajan dimension of the hypothesis class, which, again, depends on $k$ as $O(\ln k)$.
Finally, they present a trade-off between the number of oracle calls T and the size of the sample that is passed through the weak learner : at least one of them must be of order greater than $\sqrt{k}$ in order to keep a constant accuracy.

**Limitations And Societal Impact:**

Due to the theoretical nature of the paper, I do not see any potential societal impact of this work.

**Main Review:**

The paper is overall very well written and the contributions to the field seem to be strong. I have found interesting the idea of studying the algorithmic complexity in addition to the sample complexity of the proposed algorithm in order to exhibit a trade-off. As I am not an expert in boosting, I can not evaluate the impact of this paper on the litterature with high confidence.

However I think that it could have been interesting to run some experiments in order to compare Multiclass Adaboost with the other existing approaches. Moreover, concerning the trade-off I was wondering if it was somewhat optimistic : do you think that in practice, for some specific hypothesis classes, $T$ and $m^w$ must be both strictly greater $O(\sqrt{k})$ in order to keep a constant accuracy ?

**Time Spent Reviewing:**

4

---

> ### Author Response · Authors · 2021-08-10
> **Response**
>
> We thank the reviewer for reading the manuscript and for their feedback and suggestions.
>
> Regarding experiments, since the focus of this work is theoretical investigation of multi-class boosting, empirical results were beyond the scope of this work. We agree that it would indeed be interesting to conduct such experiments in future work.
>
> Regarding the reviewer’s question about the trade-off: Corollary 8 shows that for certain hypotheses classes, either number of rounds ($T$) or number of examples ($m_w$) need to be at least $\sqrt(k)$ (not both), in order to get constant accuracy.

---

### Official Review · Reviewer_6X9w · 2021-07-15

**Rating:** 5
**Confidence:** 4

**Summary:**

This paper tackles the important problem of multi-class boosting, where the base learners are multi-class learners rather than binary ones. The authors introduce a new definition for the weak learning condition in the multi-class setting and propose several theoretical results based on this new condition. The centerpiece of the paper is the multi-class Adaboost algorithm, an Adaboost-like procedure incorporating the weak learning condition.

**Limitations And Societal Impact:**

Not applicable.

**Main Review:**

Multi-class boosting has attracted a lot on attention since the early days of boosting. Several weak learning conditions as well as frameworks have been proposed, each one having strength and weaknesses. This paper introduces yet another definition of weak learning condition in the multi-class setting, tailored for cases where the number of classes is arbitrarily high. The authors make use of several tools, such as the Natarajan dimension, in order to provide several theoretical results and guarantees. The key ideas are easy to grasp and overall the contributions are interesting.

There are however several shortcoming in the current version of the paper. First of all, while the main ideas are easy to grasp, the paper in itself is not easy to read. There are several awkward formulations and typos. For instance, in Section 1, the sentence on the relation between the number of samples required by the booster and the weak learner, is repeated several times. Formulations such as base class, which refers to the class of classifiers, can be misleading in a paper about multi-class boosting. The liberal usage of hypothesis, classifier and predictor, or plurality-vote instead of majority vote (the most commonly used term), or the agnostic term, is also confusing.  Lines 178, 179 and 249 contain several typos.

Secondly, and this is my main gripe with this paper, it is unclear to me wether Algorithm 1 is derived from the theoretical contributions of the paper or wether the theoretical results are custom tailored for Algorithm 1. In the former case, I think that it would've interesting to have a (partial) derivation of the Algorithm and how all these results are combined. In the latter case, which I suspect to be the case, this severely limits the impact of the contributions.

Another important point, which follows up the previous one, is that only Algorithm 1 is studied in the theoretical discussions. I strongly think that proposing other approaches that benefit from the new WL condition would've given a better understanding of the impact of these results. Moreover, it's disappointing that there isn't a discussion on how other WL conditions and/or multi-class boosting algorithms are represented in the proposed setting, or how they compare.

Finally, while I understand the authors' motivation for including all the theoretical results in the main paper, I think that the current version of the paper is overloaded with theoretical results and definitions, making it hard to clearly understand the scope of this paper. For instance, I'm not entirely certain I understand the relation between Section 5 and Algorithm 1.

Overall, it's an interesting paper, but in my opinion, it needs to be polished a bit more.

after rebuttal : I appreciate the thorough response provided by the authors for all the points raised in my review, however I still think that the paper needs to be polished in order to deliver a clear message on its scope.

**Time Spent Reviewing:**

what's the point of this question?

---

> ### Author Response · Authors · 2021-08-10
> **Response**
>
> We thank the reviewer for reading the manuscript and for their feedback and suggestions.
>
> There seems to be a misunderstanding regarding the main contributions of the paper and, as a consequence, concerns regarding the significance of the results. We will try to alleviate the confusion by first clarifying what are the main results, and then we will address each of the specific concerns raised by the reviewer below.
>
> The review mostly revolves around Algorithm 1 (e.g., “The centerpiece of the paper is the multi-class Adaboost algorithm”), though it is not the main result of this paper. The main contribution of our work is the discrepancy between the sample complexities of the booster and weak learner, which we prove to be inherent to the task of multi-class boosting. As mentioned in the abstract, “In stark contrast, … we prove that the number of samples required by a weak learner is at least polynomial in k, exponentially more than the number of samples needed by the booster”.
>
> We wish to stress that the gap between the complexities, holds for **any** boosting algorithm, an example of which is Algorithm 1 that we analyze (Sections 3, 4). See Section 5 for the formal statement and results (Theorem 7, and Corollary 8).
>
> Overall, we are certain that the reviewer’s comments will assist us in improving the presentation of the main results, and we thank the reviewer for that. We now address specific comments raised in the review below.
>
> Regarding the semantic formulations-related comments:
>
> - We note that we have used classic definitions from the literature (base “hypotheses class” / “multi-class learning”) and we are following the standard conventions. Similarly, “hypothesis, classifier and predictor” are standard in the literature in general. However, we understand that it might be confusing, we thank the reviewer for pointing this out and we will further clarify in the final version.
>
> - “plurality-vote instead of majority vote” - These are in fact different, and there is a crucial difference between these concepts. Majority vote is in general too weak when there are  > 2 labels. Assume that the labels are {a,b,c} and that vote(a)=vote(b)=30% and vote(c)=40%. Here the *plurality* vote picks "c" whereas the majority vote is *undefined*. See Definition 1 for the formal definition of a plurality vote.
>
> - “agnostic term is also confusing” - the agnostic setting is defined in Section 2. The reviewer did not note what was not clear.
>
>
> Regarding the comments concerning Alg.1:
>
> - “...unclear whether Algorithm 1 is derived from…” - As explained above, our main results hold for any boosting algorithm; the upper bound is demonstrated via Algorithm 1 (though we note that it even holds for any ERM learner, as briefly discussed in lines 256-261), and the lower bounds hold for any booster.
>
> - “... only Algorithm 1 is studied...” - We remark that Algorithm 1 builds on classical multi-class boosting algorithms Adaboost.MR [17], but is an improvement in the sense that Alg. 1 only assumes the standard classification loss learner, and not other intricate losses.
> Moreover, albeit being simpler, Alg.1 still achieves optimal results from time and booster sample complexity perspectives.
>
> - “it's disappointing that there isn't a discussion...” - The WL we consider was introduced and thoroughly covered by Mukherjee and Schapire (2011) [13], and we provide a reduction of that condition to the classical  0-1 classification loss (Section 3). Note that Mukherjee and Schapire (2011) [13], show that this condition is optimal, and include a comprehensive comparison of it to many WL conditions. We have discussed the cost of our reduction to the classification loss, in the context of the upper bounds. However, in Section 5 we do *not* assume this particular reduction, but show a lower bound for a WL that satisfies the condition in [13], which is already known to be necessary for multi-class boosting.

---

> > ### Comment · Reviewer_6X9w · 2021-08-29
> > **Response**
> >
> > Thank you for the thorough response. I have a better understanding of the paper as a whole, however I'll have to stand by my initial evaluation of the paper. The whole point of my review is that in the current version of the paper places Algorithm 1 in a central position, thus creating the ambiguity of wether the theoretical results were created for Algorithm 1 and next extended to the general setting (especially since Sections 3 and 4 focus on Algorithm 1) or vice versa. Hence my suggestion of studying other algorithms, in order to provide a better scope for the theoretical results in Section 5. The same ambiguity goes for the agnostic setting which is not clearly introduced, however the reader is expected to understand that it is defined in equation 2. In my opinion, a better organisation of the paper should help deliver a clearer message on the contributions of this work.
> >
> > Overall a very interesting paper, but the current version does not do justice to the theoretical results introduced therein.

---

> > > ### Author Response · Authors · 2021-08-31
> > > **Response**
> > >
> > > We understand that the reviewer initially got the unintended impression that Alg. 1 is the centerpiece of the paper. However,
> > > let us again stress that Alg.1 is actually the smaller component of our main result. The lower bound is the more technically challenging, and perhaps conceptually surprising component of the main result. Let us re-iterate that this lower bound applies to $\textbf{any}$ boosting algorithm, and it implies an inherent exponential gap between the sample complexities of the booster and weak learner.
> > >
> > > We note that this main result is in fact emphasized in the abstract and introduction (and even the title of the paper). Thus,
> > > we kindly and respectfully request that the reviewer will reconsider the claim that there is an ambiguity regarding the centrality of Algorithm 1 in the main result. We hope that the reviewer will reconsider their viewpoint and scoring of the paper, after having read our responses
> > >
> > > Lastly, in any case we would like to thank the reviewer for their time and effort spent throughout the rebuttal for this ongoing discussion which will improve our overall presentation. We would also like to ask if there are any remaining issues that we have not addressed? We welcome any guidance on how to polish up the paper to the point where it could earn an unequivocal acceptance from you.

---

### Official Review · Reviewer_66dG · 2021-07-17

**Rating:** 7
**Confidence:** 2

**Summary:**

As a theoretical work, this paper considers the multi-class boosting setting. It studies the resources required for boosting,  especially how they depend on the number of classes $k$, for both the booster and weak learner. Further, it gets several interesting theoretical results.

**Limitations And Societal Impact:**

No.

**Main Review:**

First, I have to admit that am not an expert in this field, especially for the computational learning theory about boosting. But, I believe this is a solid work for multi-class boosting, especially for large class space settings.

**Time Spent Reviewing:**

3

---

> ### Author Response · Authors · 2021-08-10
> **Response**
>
> We thank the reviewer for reading the manuscript and for their feedback.

---

> > ### Comment · Reviewer_66dG · 2021-08-29
> > **Thanks for your responce**
> >
> > Dear authors,
> >
> > Thanks for your response. While I think this is a solid work in multi-class boosting, I have to admit that I am not an expert in this field and not familiar with the related work. Thus, I keep the score while decreasing the confidence.

---

### Official Review · Reviewer_DyJ5 · 2021-08-02

**Rating:** 7
**Confidence:** 4

**Summary:**

This paper focuses on the boosting approach, in a multi-class setting, with a possibly large number of classes. It lays a theoretical background on the theoretical analysis of this particular setup, giving key definitions. Then, it presents a new boosting algorithm based on this groundwork, and provides a theoretical analysis of its training behaviour. It switches its focus to the generalization guarantees of the presented method. It ends on a broad study of the multi-class boosting task. It analyses the required quantity of information needed for a boosting algorithm to perform and presents a trade-off between the complexity of the weak learner and the number of calls by the boosting algorithm.

**Ethical Concerns:**

This theoretical work has no ethical issues.

**Limitations And Societal Impact:**

The assumptions on which the theoretical work is built are clearly stated in the paper, and I can not think of a negative societal impact of this submission.

**Main Review:**

# Originality :

This paper presents a theoretical work in the multi-class boosting setup. It is a well-studied setup, as boosting is very popular, and the multiclass tasks are more and more common. However, the originality of this paper relies on the fact that it gives a theoretical framework to its algorithm and presents general and meaningful results on the sample complexity of multiclass boosting algorithms. It presents sample complexity theorems similar to Boosting : Fundation and Algorithms, but in the multiclass setup.

# Quality :

The submission presents 3 theorems that are thouroughly proven in Supplementary Materials, the theoretical analysis is strong. It does not provide any experimantal result, but as it is a theoretical work on multiclass boosting, it does not impact negatively the quality of the paper. This work is complete, as it provides an algorithm, its theoretical analysis and the resultant general theorem about the sample complexity of multiclass boosting.

# Clarity :

The submission is very dense : it provides an algorithm and a number of meaningful results in the available space. The content is well organized, starting with the theoretical background leading to the development of an algorithm. And ending on a theoretical study of this algorithm, that lead to a general result on multiclass boosting. It is clear for an informed reader that has some experience in the field, but could be a challenging read for a curious outsider that wants to have an in-depth understanding of the results. However, as the main claims are clearly provided in the introduction, the results are easily accessible.

# Significance :

This work provides a unique theoretical framework to an intuitive result : the sample complexity of multiclass boosting algorithms grows with the number of classes. This paper provide an in-depth analysis of this statement, for a specific algorithm, and in general. This is an important theoretical result, that has applications for the day-to-day use of multiclass boosting approaches. To the best of my knowledge, the sample complexity of multiclass boosting algorithm was not studied, so this work is a generalization of fundamental work on the binary version.

# Additional comments and straightforward corrections :

## Important corrections :

* Theorem 3 : $m_w = m_w(\gamma/k, \delta/T)$ is inconsistent with the proof where the assumption is made that $\epsilon = \gamma/(2k)$
* 553+ The simplification of $\epsilon_n \leq 1/2 \epsilon$ should be explained, as it is unclear as is (from eq. 59 to 60, the second term of the addition is clearly simplified in $\epsilon/2$, but the third one is left to the reader, and the calculus requires too much work to be considered trivial).
* 665 The assumption is made that $\delta^B \leq 1/2$, this means that the statement at line 640 does not lead to Theorem 7. It would if $\delta^B \geq 1/2$. However if the assumption is made that \delta^B = 1/2, it works. So there is no need for $\delta^B$ as in 664+2, you already have $P_{H, S, R}[...] < 1/2$ which is sufficient for the implications in lines 637-640. Similarly, at line 679+1

## Strainghtforward corrections (The following are not exhaustive, but only what was found during reviewing, to the best of my knowledge) :

* Last sentence of the abstract (p1 l20-22) is not that clear, especially l21-22, might want to replace "of" l21 and rephrase "the more that is demanded".
* P4 footenote 3 "for which a \gamma-realizable samples" sounds weird.
* Alg1 l6 "$y_i \neq l \in [k]$" -> "$l \in [k], l \neq y_i$" seems clearer
* Adding the notation "$\sigma$-gain" at the end of l 153 could help with a smoother understanding of the following (it is used in l 179, but never clearly introduced in the paper)

(Some lines where not labelled in the Supplementary Materials, so the comments use the last labelled line + the number of unlabelled lines)

* As the supplementary material is already quite long, it could be more reader-friendly to remind the theorems before the proofs, similarly for the lemmas in Sections C.4-6
* 463 "Then, THE corresponding"
* 499+2 & 506 change 'the Lemma' by 'Lemma \ref{lemma9}'
* 505+2 "by the the fact" remove one 'the'
* 505 + eq 35 : replace $=$ by $\leq$ as $Z_t \leq \sqrt{1-\gamma_t^2}$ and replace $(1-\gamma_t)^{1/2 - \gamma_t/4}$ -> by $(1-\gamma_t)^{1/2- \gamma_t/8}$ (no impact on the end of the proof)
* 505 + eq 36 also holds because $1+x \leq exp(x)$
* 512, 522 "a combined hypotheses" -> "a combination of hypotheses" or "the combined hypotheses"
* 566 "for the case that" sound weird
* 573,574,575,579,585 use "two" and "one" instead of "2" and "1"
* 585, 586,587  hypotheses -> hypothesis
* 617 "distribution of" -> "distribution over"
* 626+1 The lemma sounds weird : either move F, or add a comma
* 657 corresponds -> correspond
* 662 "can by" -> "can be" ?
* 673 "example was is"
* 696 'the Lemma' -> "Lemma \ref{}"
* 698 "1 hypotheses" -> "one hypothesIS"
* 715 next -> Next
* 723 "note" -> "Note"

**Time Spent Reviewing:**

13h

---

> ### Author Response · Authors · 2021-08-10
> **Response**
>
> We thank the reviewer for reading the manuscript and for their feedback and suggestions. We appreciate the comprehensive list of typos and corrections noted by the reviewer, all of which will be corrected in the final version.

---

### Decision · Program_Chairs · 2021-09-28

**Decision:**

Accept (Poster)

**Comment:**

The work is an original contribution to the theoretical study of multiclass boosting, and meets NeurIPS quality standard.

For the readers' benefit, I kindly ask the authors to take into account reviewer's 6X9w comments and clarify the scope of the contribution while preparing the camera-ready version.

**Consistency Experiment:**

NeurIPS has a long history of experimentation. In 2014, NeurIPS ran an experiment in which 10% of submissions were reviewed by two independent committees to quantify the randomness in the review process. This year, we repeated a variant of this experiment to see how the quality of the review process has changed over time.  This paper was part of the experiment and was therefore assigned to two committees (consisting of reviewers, an Area Chair, and a Senior Area Chair) that reached independent decisions.  If both committees made the same recommendation, this recommendation was followed. If a single committee recommended acceptance, the paper was accepted (with the exception of a few cases in which the other committee identified what we considered a fatal flaw, e.g., an error in a key result).

Both committees reached the same decision: **Accept (Poster)**

The other committee assigned to the paper recommended **Accept (Poster)**.  You can find the other set of reviews, along with any follow up discussion with the authors here:
https://openreview.net/forum?id=fJWmx5i5lOv